# From data to decisions: Predicting inpatient burn mortality with advanced classification models

**Yasin Sabet Kouhanjani**, **Mohammad Sattari**, **Asghar Ehteshami** *

Health Information Technology Research Center, Isfahan University of Medical Sciences, Isfahan, Iran

* ehteshami@mng.mui.ac.ir

## Abstract

### Introduction

Burn injuries are a major global health challenge. Predicting mortality in burn patients using data mining can enhance therapeutic decision-making, but the long-term reliability of such models is a critical concern for safe clinical deployment. This study aimed to develop a high-performing mortality prediction model and rigorously evaluate its temporal stability.

### Methods

This retrospective cohort study utilized data from 651 patients, comprising 93 predictive features, from the burn registry of the Injuries and Burn Subspecialized Teaching Hospital, affiliated with Isfahan University of Medical Sciences. After data preprocessing, including robust imputation using a Generalized Linear Model (GLM), five tree-based models were developed. Model performance was evaluated using a 10-fold stratified cross-validation and a rigorous temporal validation, with models trained on an initial nine-month cohort (n = 435) and tested on a subsequent nine-month cohort (n = 216).

### Results

The Gradient Boosted Trees (GBT) model demonstrated the best overall performance in cross-validation, achieving an accuracy of 93.1% (± 2.1%), an Area Under the ROC Curve (AUC) of 0.966 (± 0.010), and the lowest Brier Score of 0.060 (± 0.017), indicating superior discrimination and calibration. The most influential predictors were the Abbreviated Burn Severity Index (ABSI), Total Burn Surface Area (TBSA), and the percentage of third-degree burns. Crucially, the temporal validation revealed strong model stability, with the GBT model's AUC only decreasing slightly to 0.948. However, a clinically significant drop in sensitivity was observed (from 78.1% to 68.3%).

**Data availability statement:** This study was a secondary analysis of data extracted from the burn registry of the Injuries and Burn Subspecialized Teaching Hospital, affiliated with Isfahan University of Medical Sciences. Although the dataset was de-identified prior to analysis, it remains subject to national data protection laws and institutional privacy regulations. Due to restrictions imposed by the Ethics Committee and data owner, the dataset cannot be shared publicly. However, access to a minimal de-identified dataset may be granted to qualified researchers upon reasonable request. Data access requests should be submitted formally to the Isfahan University of Medical Sciences Ethics Committee at resrec@mui. ac.ir. The full RapidMiner processes used for model development and validation in this study are publicly available in the Zenodo repository under DOI: 10.5281/zenodo.15639850.

**Funding:** The author(s) received no specific funding for this work.

**Competing interests:** The authors have declared that no competing interests exist.

## Conclusion

Tree-based models, particularly GBT, are powerful and accurate tools for predicting burn mortality. While our model demonstrated strong temporal stability, the observed decrease in sensitivity highlights that even robust models are subject to performance shifts over time. This underscores the vital importance of implementing a governance framework, including continuous monitoring and periodic recalibration, to ensure the sustained safety and efficacy of predictive models in clinical practice.

## Introduction

Burn injuries are among the most common and severe medical conditions globally, leading not only to physical trauma but also to substantial psychological, social, and economic consequences [1]. According to the World Health Organization (WHO), burn injuries result in approximately 180,000 deaths annually, with the majority occurring in low- and middle-income countries [2]. In Iran, data from 2017 show that more than 35,000 patients were hospitalized due to burns, highlighting the significant national burden and emphasizing challenges in burn prevention and management within the healthcare system [3]. Burn injuries are also associated with long-term physical and psychological complications, high medical costs, and income loss [4]. Despite significant advances in burn care, the complex physiological responses to burns continue to pose major challenges to improving patient outcomes [5]. Moreover, burn-related complications can further hinder recovery. One such complication is gastrointestinal-origin abdominal pain, which has been identified as a prevalent yet often overlooked post-burn issue that requires dedicated clinical attention [6]. As healthcare systems increasingly adopt digital technologies, access to vast amounts of health-related data has opened new possibilities for optimizing patient care [7]. It is estimated that over a patient's lifetime, approximately one million gigabytes of health data are generated, with this volume doubling every 2–5 years [8]. Analyzing such high-dimensional datasets exceeds the capabilities of traditional statistical methods and requires advanced techniques such as data mining [9]. Data mining, which incorporates statistical modeling, machine learning, and artificial intelligence, enables the extraction of hidden patterns and meaningful insights from large datasets [10,11]. In medicine, these tools have been applied to support prediction, diagnosis, and treatment decision-making [12].

Recent studies have demonstrated the value of data mining in burn care. For example, Ahmadi et al. (2021) used data mining to examine factors influencing the delay between burn injury and treatment initiation [13]. Yang et al. (2023) employed gradient boosting to predict inhalation injury with high diagnostic accuracy [14]. Aydin et al. (2023) applied machine learning to classify esophageal burns in children following caustic ingestion [15]. Yazici et al. (2023) evaluated various machine learning algorithms for predicting burn-related mortality [16]. In Iran, Ehteshami et al. (2024) analyzed data from the Injuries and Burn Subspecialized Teaching Hospital affiliated with Isfahan University of Medical Sciences using nine features to compare several algorithms for mortality prediction [17].

However, the clinical applicability of such models often faces two critical challenges. First, the generalizability of results from such models may be limited due to dataset-specific characteristics [18]. Second, their reported performance, may not reflect the model's stability over time in a dynamic clinical environment, a challenge known as 'concept drift' [19]. To address these gaps, the present study aims to identify key predictors of burn mortality and to develop robust predictive models using multiple data mining techniques, including Decision Trees (DT), Random Forest (RF), Gradient Boosted Trees (GBT), Random Tree (RT), and Decision Stump (DS). Unlike traditional prognostic models such as the Abbreviated Burn Severity Index (ABSI) and Baux scores, which rely on a limited number of predefined clinical variables, our approach utilizes advanced tree-based machine learning algorithms capable of capturing complex, non-linear interactions across a broader set of input features [20,21]. In contrast to prior machine learning studies, this research evaluates a diverse range of classifiers using a unique and feature-rich dataset extracted from a specialized burn registry. While the study by Ehteshami et al. utilized a limited set of features from the same hospital [17], our research is based on a larger, more comprehensive dataset derived from the hospital's burn registry, incorporating a wider range of variables and more recent patient records. While the first part of our approach addresses these challenges by enhancing potential accuracy and generalizability, a second and more critical objective of this study is to conduct a rigorous temporal validation. This step allows us to directly assess our model's stability and evaluate its robustness against potential concept drift, thereby establishing greater confidence in its applicability for real-world clinical deployment.

## Methods

### Proposed methodology

For this research, Microsoft Excel 2021 was employed for initial data examination and part of the preprocessing tasks, while RapidMiner software (version 9.10.000) was used for advanced data preprocessing and modeling. The methodology applied was based on the **CRISP-DM framework** (Cross Industry Standard Process for Data Mining), which is recognized as one of the most robust methodologies for implementing and executing data mining projects [22]. The CRISP-DM framework includes six key phases: System understanding, Data Understanding, Data preprocessing, Modeling, Evaluation, and Deployment. This study covers the first five phases of this framework, from understanding the problem to the final evaluation of the models.

### Dataset description

This retrospective cohort study utilized data from the **burn registry of the Injuries and Burn Subspecialized Teaching Hospital**, affiliated with Isfahan University of Medical Sciences, covering the period from **12/04/2022 to 08/10/2023**. Each patient is represented by a single observation (row) in the dataset; no longitudinal data were used. Following an initial assessment, 11 records with a missing final outcome were excluded from the dataset. Thus, the final analytical dataset consisted of a cohort of 651 patients (144 deceased and 507 surviving), for whom 93 predictive features were used for model development, a complete list of which is provided in S1 Table. The dataset was retrospectively accessed for research purposes between **21/06/2024 and 21/09/2024**.

### Data preprocessing

Prior to applying classification techniques, the following preprocessing steps were performed:

**Initial feature engineering and selection.** The initial dataset consisted of 720 features. To create a relevant and manageable feature set for mortality prediction at the time of admission, an initial selection was performed. Features not recorded during the emergency phase or unrelated to the study's objective were excluded, resulting in 124 features. Subsequently, 24 redundant or similar features were consolidated, and 6 composite features were expanded into 38 new binary variables using a one-hot encoding approach. After removing 16 features with highly imbalanced classes (where over 95% of values belonged to a single class, for instance, a specific comorbidity present in less than 5% of patients),

121 candidate features remained. Subsequently, a further data reduction step was performed to handle highly incomplete features. From the 121 candidate features, 28 were removed because they had more than 85% missing values. This threshold was chosen as a pragmatic balance to remove only the most sparsely populated features, which could introduce noise, while retaining a comprehensive set of potentially valuable clinical variables. This process resulted in a final set of 93 features used for model development.

**Handling missing values.** Missing data remained a challenge in the final 93-feature set. An analysis revealed that 79 of the 93 features (approximately 85%) contained at least some missing values; however, the degree of missingness was generally low, with the majority of these affected features having less than 20% missing data. The specific percentage of missing values for each individual feature is detailed in S1 Table.

To handle this, model-based imputation was employed using the 'Impute Missing Values' operator within RapidMiner. A Generalized Linear Model (GLM) was used as the learner within this operator. This advanced imputation method was chosen over simpler techniques (e.g., mean imputation) because it is capable of capturing complex relationships between variables, and our sensitivity analysis confirmed it yielded a slight improvement in overall model performance. The GLM was configured with its family and solver parameters set to 'AUTO', allowing the software to select the optimal distribution and optimization algorithm for each feature. Key parameters for the GLM also included the use of regularization and data standardization to ensure robust model fitting.

Crucially, to prevent data leakage from the outcome into the feature set, the outcome variable (label) was excluded from the imputation process. This ensures that information about the final patient outcome did not influence the imputation of predictor variables. Furthermore, to prevent overly optimistic performance estimates, the entire imputation process was nested within the model training and evaluation procedure. This ensures that for each evaluation run, the imputation model was built using only information from the training data for that specific run.

## Model development and evaluation

The predictive models were developed using the final set of 93 features as independent variables to predict the binary 'Final outcome' variable (deceased vs. survived) as the target.

**Model Development and Hyperparameter Tuning.** The study focused on tree-based ensemble models due to their strong performance on tabular data, their inherent ability to capture complex non-linear relationships without extensive feature engineering, and their relatively high interpretability compared to 'black-box' models. While other model families like logistic regression are valuable, the objective here was a deep evaluation of this specific class of powerful algorithms rather than a broad, shallow comparison across multiple families.

Five tree-based algorithms were implemented in RapidMiner (version 9.10.000): Gradient Boosted Trees (GBT), Decision Tree (DT), Random Forest (RF), Decision Stump (DS), and Random Tree (RT). To select optimal hyperparameters and simultaneously obtain an unbiased estimate of each model's final performance, a nested validation procedure was implemented. The outer loop utilized the 'Optimize Parameters (Grid)' operator to systematically search for the optimal combination of hyperparameters for each algorithm. The parameter grids and search ranges are detailed in S2 Table. To robustly evaluate the performance of each hyperparameter combination within this search, a complete 10-fold cross-validation was executed as the inner loop of the process. This inner validation was configured to use shuffled sampling to partition the data. The performance metric used to guide the optimization process and select the best hyperparameter set was overall Accuracy. The final performance metrics reported in this study represent the performance of the models configured with these optimal hyperparameters, as evaluated by this robust nested procedure. The optimal hyperparameters selected through this process for the primary models are presented in Table 1.

**Model performance evaluation.** The performance of the finalized models was rigorously assessed using a 10-fold cross-validation process with stratified sampling. Stratified sampling was chosen to ensure that the proportion of outcome classes (deceased vs. survived) was maintained across all folds, thereby providing a more stable and reliable estimate

**Table 1. Optimal hyperparameters selected for each model.**

| Model | Hyperparameter | Optimal Value |
|---|---|---|
| GBT | number_of_trees | 20 |
| | maximal_depth | 5 |
| | learning_rate | 0.1 |
| DT | criterion | information_gain |
| | maximal_depth | 3 |
| | minimal_size_for_split | 2 |
| RF | number_of_trees | 20 |
| | maximal_depth | 50 |
| | criterion | gini_index |
| | voting_strategy | confidence vote |
| DS | criterion | gain_ratio |
| | minimal_leaf_size | 1 |
| RT | criterion | gain_ratio |
| | maximal_depth | 40 |
| | minimal_size_for_split | 10 |
| | number_of_prepruning_alternatives | 7 |

of performance. For all classification algorithms, the default decision threshold of 0.5 was applied to convert predicted probabilities into binary class labels (Deceased vs. Survived).

Model performance was quantified using several standard metrics: Accuracy, Precision (Positive Predictive Value), Sensitivity (Recall), Specificity, and F1-Score. Additionally, the Area Under the ROC Curve (AUC) was used to assess the overall discriminative ability. To evaluate model calibration —a measure of how well the predicted probabilities match actual outcomes—the Brier Score was also calculated. A lower Brier Score indicates better calibration and more reliable probability estimates, a crucial feature for clinical risk stratification tools.

To determine if the observed differences in performance metrics between models were statistically significant, paired t-tests were conducted on the metric scores obtained from the 10-fold cross-validation. A p-value of less than 0.05 was considered statistically significant. This study adheres to the Transparent Reporting of a multivariable prediction model for Individual Prognosis or Diagnosis (TRIPOD) statement; the completed checklist is provided in S1 Checklist.

**Assessment of feature importance.** To identify the most influential predictors from the final feature set, a two-stage feature importance analysis was conducted. First, for an initial exploratory overview, the importance of each feature was assessed using several standard weighting methods within RapidMiner (e.g., Gini Index, Information Gain); the full results of this exploratory analysis are available in the Supporting Information (S3 Table).

Second, for the definitive analysis, model-based feature importance scores were extracted from the final trained models. A detailed visualization and interpretation of these scores, focusing on the best-performing model, is presented in the Results section.

**Temporal validation and sensitivity analyses.** To assess model stability—defined here as the consistency of model performance on future data from the same clinical environment—and to investigate the potential for concept drift, a rigorous temporal validation was performed. The dataset was divided into two non-overlapping chronological cohorts based on each patient's admission date, ensuring that each unique patient belonged to only one cohort and thus preventing data leakage. The training cohort consisted of all patients admitted during the first nine months of the study period (n = 435), while the temporal test cohort comprised all patients from the subsequent nine months (n = 216). It is important to note that this temporal validation assesses the model's robustness to shifts in the patient population at a

single center over time, and is distinct from external validation, which assesses a model's generalizability across different institutions.

For this temporal analysis, the evaluation focused on the two primary ensemble models: Gradient Boosted Trees (GBT) and Random Forest (RF). Each model was configured using the optimal hyperparameters identified during the nested validation procedure, as presented in Table 1. The finalized models were trained on the entire training cohort, and their performance was subsequently evaluated once on the entire temporal test cohort.

Furthermore, to investigate the robustness of our findings to methodological choices, three sensitivity analyses were conducted on the top-performing models (GBT and RF). The first analysis compared the performance of our primary imputation method (GLM) against a traditional Mean/Fixed Value Imputation. The second analysis assessed the impact of variable format by comparing the performance of models built with Continuous variables against those using categorized variables. For this comparison, the categorical variables were generated by discretizing the original Continuous features based on standard medical thresholds and conventional classification criteria. A third sensitivity analysis was performed to evaluate the model's reliance on the pre-engineered ABSI score by re-evaluating the models after this feature was excluded from the dataset.

### Ethical considerations

The study protocol was reviewed and approved by the Ethics Committee of Isfahan University of Medical Sciences (Approval ID: IR.MUI.NUREMA.REC.1403.048), and all methods were performed in accordance with the relevant guidelines and regulations of the Declaration of Helsinki. This research involved a secondary analysis of fully de-identified registry data. The Ethics Committee of Isfahan University of Medical Sciences explicitly waived the need for individual informed consent due to the retrospective nature of the study and the use of anonymized data, which posed no risk to patient privacy. Therefore, no written or verbal consent was obtained from participants or their legal guardians.

## Results

### Study population

Of the 662 burn inpatients initially identified, 11 were excluded due to missing final outcome data. The final analysis was performed on 651 patients, of whom 144 (22.1%) died during hospitalization and 507 (77.9%) survived. The flow of patient selection for the study is illustrated in Fig 1. Detailed baseline demographic and clinical features of the study population are provided in S1 Table, while a summary of the feature categories and variable types used for model development is presented in Table 2.

### Main model performance

The performance metrics for the five finalized models, evaluated using 10-fold cross-validation, are presented in Table 3. The GBT model demonstrated the highest performance across most metrics, achieving an accuracy of 93.06% and the best Brier Score of 0.060. The RF model also showed strong performance, with a comparable accuracy of 92.91% and an AUC of 0.961. The receiver operating characteristic (ROC) curves for the primary models are shown in Fig 2, visually illustrating the superior discriminative ability of the GBT and RF models. The aggregated confusion matrices for each primary model, summarizing the cumulative predictions across all cross-validation folds, are presented in Supporting information (S4 Table).

A closer examination of the performance metrics for the top-performing GBT model reveals important clinical trade-offs. The model achieved a high specificity of 97.5%, indicating it correctly identified the vast majority of patients who would survive. However, its sensitivity (recall) was 78.1%, meaning it correctly identified approximately four out of five patients who would ultimately die. The Precision (PPV) of 89.7% suggests that when the model predicted a patient would die, that prediction was correct nearly 90% of the time. These class-specific metrics are crucial for understanding the model's real-world utility beyond overall accuracy.

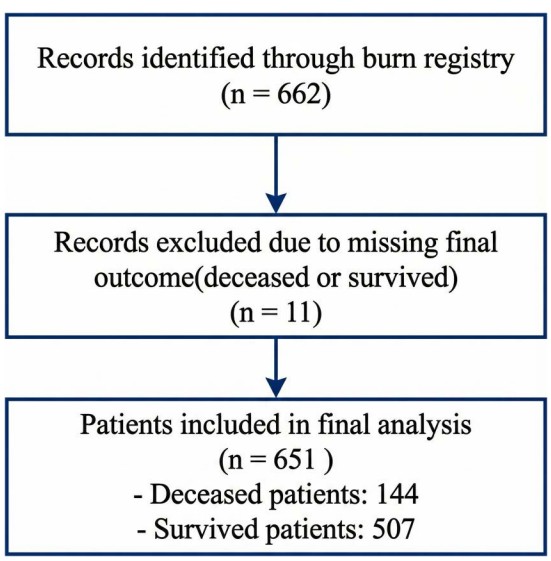

**Fig 1. Flow diagram of patient selection.** The flowchart illustrates the process of including and excluding patients for the study. A total of 662 burn inpatients were initially identified from the registry, of whom 11 were excluded due to missing final outcome data, resulting in a final analytical cohort of 651 patients.

**Table 2. Summary of feature categories and variables used for model development.**

| Feature Category (Based on Registry Groups) | No. of Variables | Key Examples | Missing Data Handling |
|---|---|---|---|
| **Demographic Characteristics** | 12 | Age, Gender, BMI, Education, Insurance Status | GLM Imputation |
| **Medical History** | 17 | Comorbidities (Diabetes, Cardiac, etc.), Smoking, Addiction | GLM Imputation |
| **Admission & Referral Data** | 7 | Transport Mode, Braden/Wells/Morse Scores, Tetanus History | GLM Imputation |
| **Burn Characteristics & Injury Details** | 23 | TBSA, Burn Degree, Burn Agent, Burn Site, Inhalation Injury | GLM Imputation |
| **Care & Interventions** | 22 | Pre-hospital Care, ER Procedures (Intubation, Dressing), Post-ER Status | GLM Imputation |
| **Paraclinical Indicators** | 11 | HGB, WBC, Electrolytes (Na, K), Albumin, Creatinine | GLM Imputation |
| **Composite Severity Score** | 1 | Abbreviated Burn Severity Index (ABSI) | GLM Imputation |
| **Total Predictors** | 93 | | |
| **Outcome (Target)** | 1 | Final Outcome (Deceased/ Survived) | Excluded (Complete Cases) |

**Note:** Detailed descriptive statistics, including means, standard deviations, and frequencies for all 93 features, are provided in S1 Table in the Supporting Information. GLM: Generalized Linear Model. The 'Final Outcome' variable served as the target label for classification and was excluded from the predictor set to prevent data leakage.

## Statistical comparison of models

To assess whether the observed performance differences between models were statistically significant, paired t-tests were conducted on the metric scores from the 10-fold cross-validation. The analysis revealed no statistically significant difference between the two top-performing models, GBT and RF, across the key metrics of Accuracy (p = 0.896), AUC (p = 0.393), and Brier Score (p = 0.620). However, both GBT and RF demonstrated a statistically significant superiority over the other models (DT, DS, and RT) on most metrics, particularly AUC (p < 0.05 for all comparisons). The full pairwise comparison results for all metrics are provided in the Supporting information (S5–S7 Tables).

**Table 3. Performance metrics of the final models.**

| Model | Accuracy | AUC | Precision (PPV) | Sensitivity (Recall) | Specificity | F1-Score | Brier Score |
|---|---|---|---|---|---|---|---|
| GBT | 93.06%±2.12% | 0.966±0.010 | 89.65%±8.09% | 78.12%±10.18% | 97.53%±1.79% | 0.827±0.043 | 0.060±0.017 |
| DT | 91.40%±2.22% | 0.901±0.044 | 87.12%±11.96% | 72.49%±12.45% | 97.01%±2.68% | 0.777±0.069 | 0.073±0.019 |
| RF | 92.91%±2.81% | 0.961±0.012 | 89.43%±6.61% | 76.39%±12.85% | 97.49%±1.57% | 0.819±0.070 | 0.063±0.011 |
| DS | 87.93%±4.18% | 0.734±0.066 | 96.33%±6.18% | 47.39%±13.31% | 99.41%±0.95% | 0.624±0.128 | 0.106±0.031 |
| RT | 86.56%±4.73% | 0.769±0.109 | 74.77%±11.65% | 57.64%±17.79% | 94.59%±3.81% | 0.651±0.123 | 0.110±0.030 |

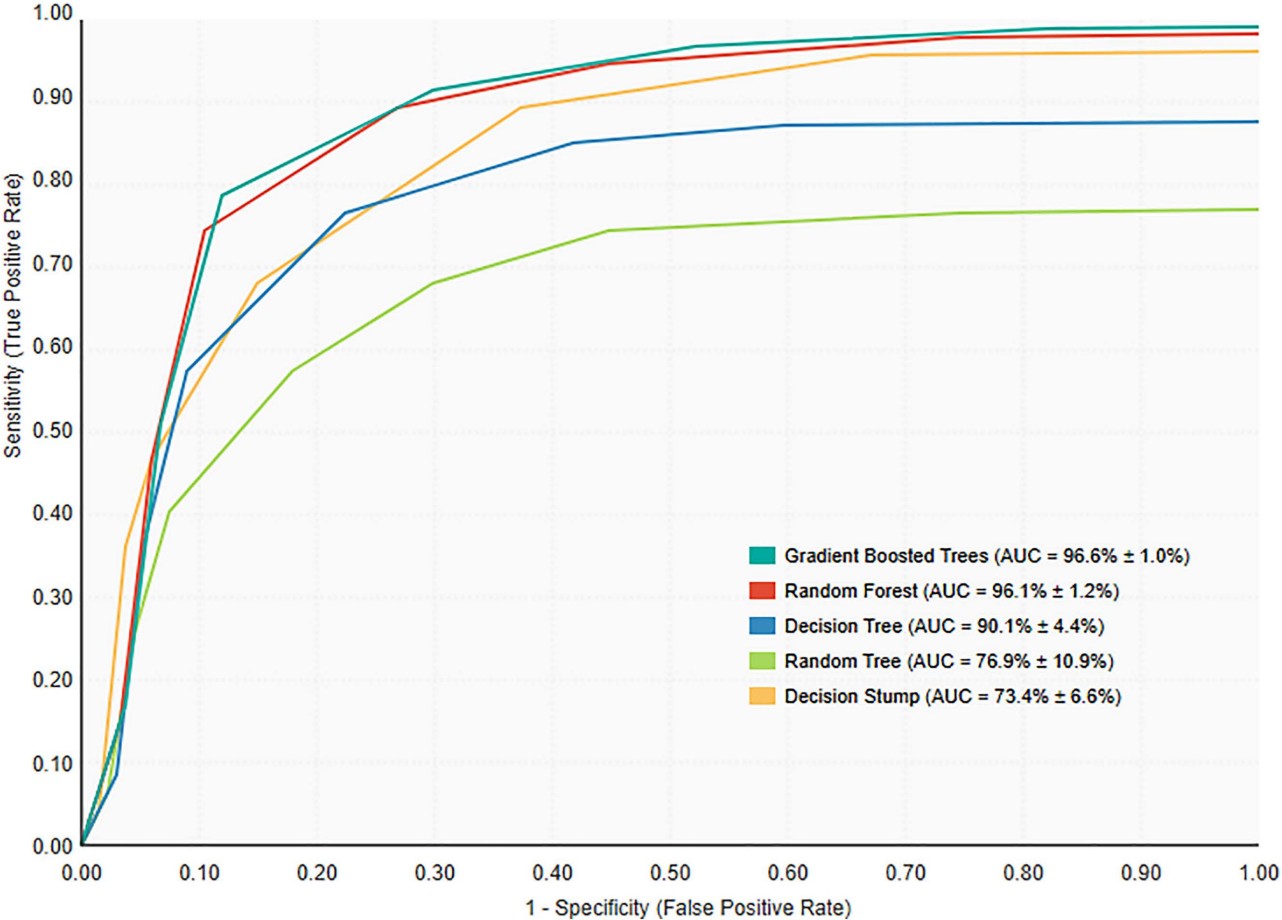

**Fig 2. Receiver Operating Characteristic (ROC) curves for the primary predictive models.** The curves illustrate the diagnostic ability of the Gradient Boosted Trees (GBT), Random Forest (RF), Decision Tree (DT), Decision Stump (DS), and Random Tree (RT) models. The Area Under the Curve (AUC) for the GBT model (0.966) and the RF model (0.961) demonstrates superior discrimination compared to the other models.

## Feature importance analysis

As shown in Table 3, the GBT model demonstrated the highest overall performance. Therefore, a deeper analysis of its feature importance was conducted to understand the key drivers of its predictions. Notably, the Abbreviated Burn Severity Index (ABSI) Score, the percentage of third-degree burns, and Total Burn Surface Area (TBSA) were identified as the most critical predictors. The top 20 most influential features, as determined by the GBT model's internal weights, are visualized in Fig 3.

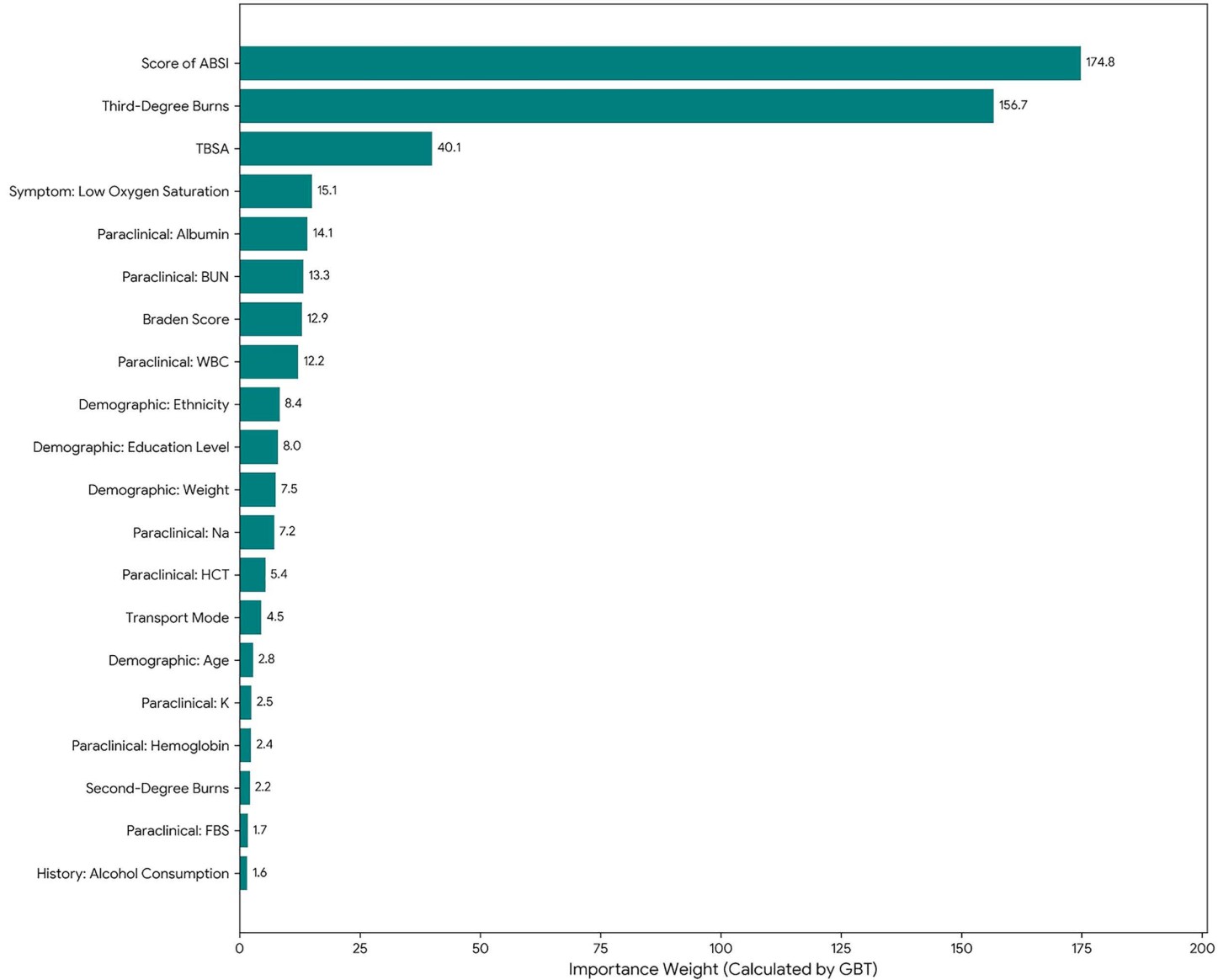

**Fig 3. Top 20 most influential features for mortality prediction identified by the Gradient Boosted Trees (GBT) model.** The horizontal bar chart displays the relative importance of the top predictors. The Abbreviated Burn Severity Index (ABSI) Score, percentage of third-degree burns, and Total Burn Surface Area (TBSA) were identified as the most critical features driving the model's predictions.

## Temporal validation

To assess model stability, the top-performing models were subjected to a temporal validation pipeline, as illustrated in Fig 4. As shown in Table 4 and visualized in Fig 5, both the GBT and RF models demonstrated strong temporal stability, retaining high overall accuracy and discriminative ability. The GBT model's AUC, for instance, saw only a minor drop from 0.966 to 0.948. The most notable change was a decrease in sensitivity, particularly for the GBT model (from 78.1% to 68.3%), while specificity remained excellent.

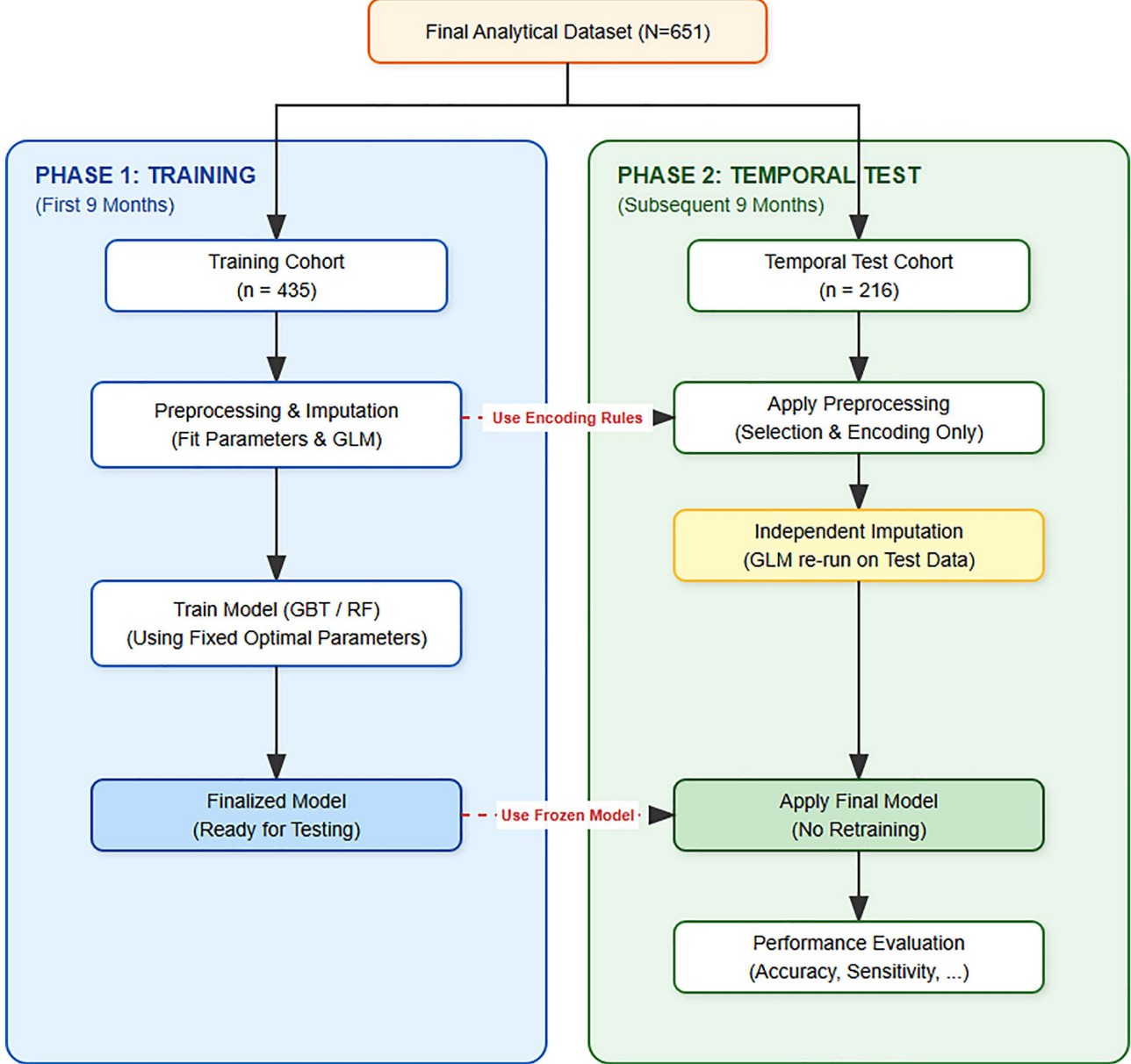

**Fig 4. Schematic diagram of the model development and temporal validation pipeline.** The dataset was split chronologically. To prevent data leakage, the final classification model was trained on the training cohort (first 9 months) using the fixed optimal hyperparameters identified during the development phase. For the temporal test cohort (subsequent 9 months), preprocessing steps—including independent GLM imputation to address concept drift—were performed, after which the frozen classifier was applied to predict outcomes without retraining.

To investigate the potential cause of this shift in performance, the baseline characteristics of the patients in the training and temporal test cohorts were compared. The results of this statistical comparison are presented in Table 5. Notably, a statistically significant difference was observed in the mean percentage of third-degree burns between the two cohorts (p<0.001), while other key characteristics such as age and TBSA remained stable.

**Table 4. Comparison of model performance under cross-validation vs. temporal validation.**

| Model | Accuracy | AUC | Precision (PPV) | Sensitivity (Recall) | Specificity | F1-Score | Brier Score |
|---|---|---|---|---|---|---|---|
| GBT (Cross-Validation) | 93.06%±2.12% | 0.966±0.010 | 89.65%±8.09% | 78.12%±10.18% | 97.53%±1.79% | 0.827±0.043 | 0.060±0.017 |
| RF (Cross-Validation) | 92.91%±2.81% | 0.961±0.012 | 89.43%±6.61% | 76.39%±12.85% | 97.49%±1.57% | 0.819±0.070 | 0.063±0.011 |
| GBT (Temporal) | 92.13% (87.8% − 95.0%) | 0.948 | 87.50% (71.9% − 95.0%) | 68.29% (53.0% − 80.4%) | 97.71% (94.0% − 99.1%) | 76.71% | 0.062±0.000 |
| RF (Temporal) | 91.20% (86.7% − 94.4%) | 0.936 | 84.38% (68.2% − 93.1%) | 65.85% (50.6% − 78.4%) | 97.14% (93.3% − 98.8%) | 73.97% | 0.067±0.143 |

**Note:** [a]Values represent Mean ± Standard Deviation (SD) derived from 10-fold stratified cross-validation. [b]Values represent the point estimate followed by the 95% Confidence Interval (CI) in parentheses. CIs for Accuracy, Sensitivity, Specificity, and Precision were calculated using the Wilson Score Interval method (n=216). AUC, F1-Score, and Brier Score are reported as point estimates.

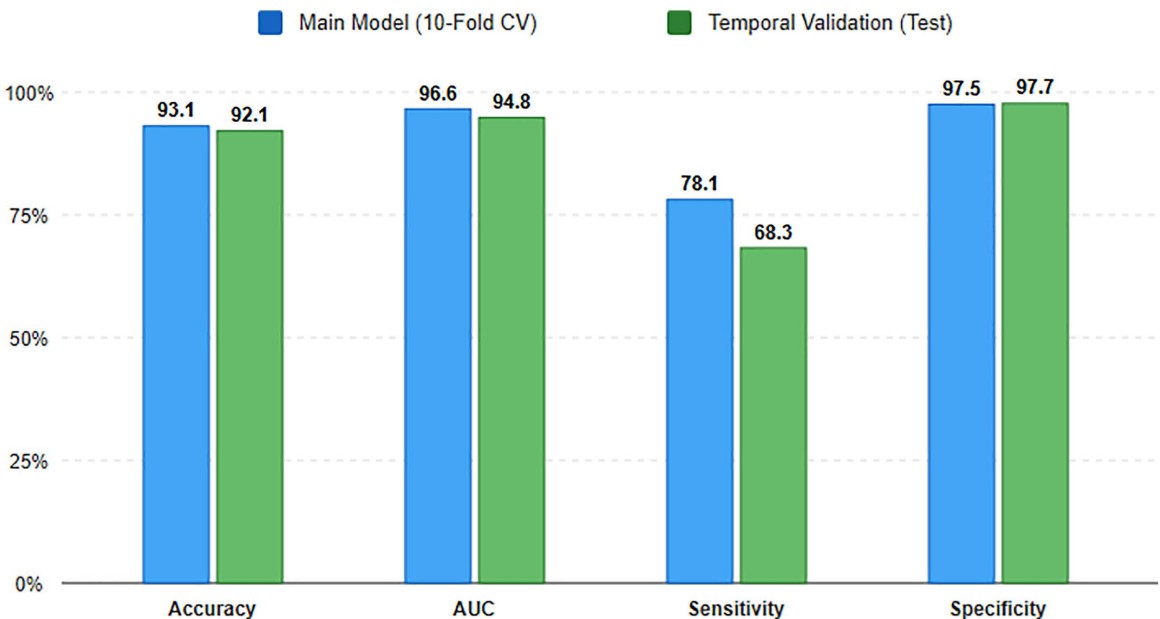

**Fig 5. Visual comparison of performance metrics for the best-performing model (Gradient Boosted Trees) across development and temporal validation phases.** The blue bars represent the mean performance of the Main Model (derived from 10-fold cross-validation), while green bars show the performance during the Temporal Validation phase. The chart highlights the model's stability in terms of Accuracy, AUC, and Specificity, while illustrating the impact of temporal concept drift through a notable reduction in Sensitivity. (Numerical details with confidence intervals are provided in Table 4).

## Sensitivity analyses

The results of the three sensitivity analyses, presented in Table 6, confirmed the model's general robustness to different methodological choices. First, a comparison of imputation methods showed that while the baseline GBT model using GLM imputation had slightly higher performance point estimates than the model using Mean Imputation, this difference was not statistically significant (p = 0.404 for AUC). Second, the analysis of variable format revealed a similar pattern, where using continuous variables did not provide a statistically significant advantage over their categorized counterparts (p = 0.638 for GBT AUC); this suggests that the discretization process may lead to some loss of predictive information, although

**Table 5. Baseline demographic and clinical characteristics of patients in the training and testing- temporal validation.**

| Characteristic | Training Set (First 9 months) (n = 435) | Temporal Test Set (Second 9 months) (n = 216) | P-Value |
|---|---|---|---|
| Age (years), Mean±SD | 33.63±19.48 | 32.40±18.45 | 0.432 |
| TBSA (%), Mean±SD | 39.25±25.04 | 36.23±24.87 | 0.146 |
| Third-Degree Burns (%), Mean±SD | 28.88±25.02 | 20.41±23.36 | **<0.001** |
| Gender (Male), n (%) | 311 (71.49%) | 164 (75.92%) | 0.269 |
| Inhalation Injury (Yes), n (%) | 294 (67.58%) | 154 (71.29%) | 0.383 |
| Final Outcome (Deceased), n (%) | 103 (23.67%) | 41 (18.98%) | 0.208 |

**Table 6. Sensitivity analysis of top models (GBT and RF) under different methodological conditions.**

| Model | Methodological Condition | Accuracy | AUC | Precision (PPV) | Sensitivity (Recall) | Specificity | F1-Score | Brier Score |
|---|---|---|---|---|---|---|---|---|
| GBT | Baseline Model (Continuous + GLM) | 93.06%±2.12% | 0.966±0.010 | 89.65%±8.09% | 78.12%±10.18% | 97.53%±1.79% | 0.827±0.043 | 0.060±0.017 |
| | Mean/Fixed Value Imputation | 92.75%±2.31% | 0.961±0.014 | 87.19%±10.31% | 79.38%±9.18% | 96.77%±2.53% | 0.823±0.046 | 0.064±0.016 |
| | Categorical Variables + GLM | 92.30%±2.28% | 0.962±0.022 | 89.98%±10.28% | 72.54%±9.65% | 97.95%±2.01% | 0.796±0.066 | 0.066±0.021 |
| | Model without ABSI score | 91.55%±3.41% | 0.945±0.041 | 81.21%±12.49% | 79.53%±13.88% | 94.96%±3.09% | 79.57%±10.61% | 0.068±0.018 |
| RF | Baseline Model (Continuous + GLM) | 92.91%±2.81% | 0.961±0.012 | 89.43%±6.61% | 76.39%±12.85% | 97.49%±1.57% | 0.819±0.070 | 0.063±0.011 |
| | Mean/Fixed Value Imputation | 92.50%±2.60% | 0.967±0.015 | 89.83%±10.39% | 73.61%±9.15% | 97.68%±2.98% | 0.808±0.055 | 0.065±0.013 |
| | Categorical Variables + GLM | 92.46%±2.31% | 0.959±0.017 | 88.52%±9.78% | 75.00%±11.17% | 97.30%±2.24% | 0.808±0.046 | 0.067±0.017 |
| | Model without ABSI score | 91.09%±3.90% | 0.950±0.045 | 83.60%±13.39% | 74.59%±17.58% | 95.92%±3.35% | 77.65%±11.87% | 0.069±0.020 |

the overall high performance is maintained. Finally, the analysis evaluating the impact of the ABSI score confirmed that its exclusion also did not result in a statistically significant drop in the GBT model's performance (p = 0.53 for GBT AUC). These findings collectively reinforce the model's robustness (see S5-S8 Tables for full pairwise statistical comparisons).

## Discussion

In this study, we developed and validated several machine learning models to predict inpatient mortality in burn patients using a comprehensive registry dataset. Our primary analysis identified the GBT model as the most effective algorithm. While the GBT and RF models showed comparable performance in terms of accuracy, GBT's superiority was evident in its higher AUC (0.966) and lower Brier Score (0.060). The lower Brier score, in particular, suggests that GBT provides more reliable and well-calibrated probability estimates, a crucial feature for clinical decision support tools intended for risk stratification [23]. Interestingly, the statistical comparison did not show a significant difference for the Brier Score between the top models (p = 0.620), suggesting they both reached a high level of calibration.

A key finding of our study was the notable temporal stability of the GBT model. Unlike many clinical prediction models that can suffer from significant performance degradation over time due to concept drift [19], our model retained high overall accuracy and discriminative ability when tested on a future, unseen patient cohort from a subsequent nine-month

period. This robustness suggests that the model learned generalizable, underlying patterns from the initial training data rather than spurious, time-sensitive correlations.

However, the temporal validation also revealed a critical clinical trade-off. The primary impact of the minor concept drift present in the data was a notable decrease in sensitivity (from 78.1% to 68.3%). This implies that while the model remains highly reliable in identifying patients who will survive (stable high specificity), its ability to detect high-risk patients who will ultimately die may diminish over time. This finding was likely driven by the observed shift in the patient population, specifically the statistically significant decrease in the percentage of third-degree burns in the temporal test cohort (Table 5).

The high performance of our GBT model in cross-validation (AUC = 0.966) is consistent with, and in some cases exceeds, the performance reported in other studies that have applied machine learning to predict burn mortality. Our findings align with those of Yazıcı et al. [16], who also identified tree-based ensembles as powerful predictors in this domain. However, a primary contribution that distinguishes our study is the rigorous temporal validation. While many studies demonstrate high predictive accuracy, the assessment of model stability over time is often overlooked. By confirming our model's robustness, our work extends beyond simply applying machine learning, providing crucial evidence for the potential long-term reliability of such models in a real-world clinical setting.

Our feature importance analysis (Fig 3) confirmed that the model's predictions were driven by clinically established risk factors, with the Abbreviated Burn Severity Index (ABSI), percentage of third-degree burns, and Total Burn Surface Area (TBSA) ranking as the most influential predictors [16,20,21]. This alignment with existing clinical knowledge increases confidence in the model's validity and suggests that it did not identify any entirely novel or unexpected predictors. Furthermore, the model's robustness was confirmed by a series of sensitivity analyses: neither the choice of imputation method nor the variable format resulted in a statistically significant difference in performance, and the model's predictive power remained strong even after the pre-engineered ABSI score was removed (see Table 6).

## Clinical implications

A highly accurate and stable model like the GBT could be a valuable asset for early risk stratification in burn units. However, our findings on decreased sensitivity carry a critical warning for clinical implementation. A model that becomes less sensitive over time could lead to false reassurances and missed opportunities for early, aggressive intervention in patients who are incorrectly flagged as low-risk. This underscores that even for robust models, periodic performance monitoring and recalibration are essential components of a responsible clinical deployment strategy to ensure sustained patient safety.

## Strengths and limitations

This study has several strengths, including the use of a comprehensive clinical registry, a rigorous nested validation procedure for hyperparameter tuning, and the inclusion of a temporal validation analysis. Nevertheless, several limitations should be acknowledged. First, as the data were collected from a single medical center, the external generalizability of our findings to other populations may be limited. However, this single-center design also provided a controlled environment to specifically test for temporal stability, which strengthens the internal validity of this key analysis. It is critical to note that our temporal validation is not a substitute for external validation. Second, the sample size (n = 651), while substantial for a single-center study, may still be insufficient for developing models that are robustly generalizable at a national or international level. Additionally, given the ratio of features to patient records, the risk of overfitting, while mitigated by cross-validation, cannot be entirely dismissed. Third, our analysis was intentionally focused on tree-based models to allow for a deep evaluation of this specific class of algorithms; consequently, other model families such as logistic regression or more complex approaches like deep neural networks were not explored. While our model-based feature importance provided robust insights, we acknowledge that more advanced interpretability techniques like SHAP analysis, which can offer deeper explanations of individual predictions, could not be implemented due to technical limitations of our analytical software. These technical constraints also influenced the temporal validation pipeline; specifically, the GLM imputation

model could not be directly serialized for the test set. Consequently, imputation for the temporal cohort was performed independently, an approach we consider appropriate given the observed concept drift to respect the specific distributional characteristics of the new time period. Furthermore, regarding statistical evaluation, we acknowledge that cross-validation folds are not strictly independent; thus, we have emphasized the confidence intervals from the independent temporal validation set and relied on Brier Scores for calibration assessment in the absence of retained probability logs. Finally, as a retrospective study based on registry data, there is inherent potential for unmeasured confounding variables and data quality issues, although extensive preprocessing and robust imputation methods were employed to mitigate this.

## Conclusion

This study demonstrated that tree-based machine learning algorithms, particularly the GBT model, are powerful tools for predicting inpatient mortality in burn patients with high accuracy. The model's validity was reinforced by the identification of clinically established risk factors—such as the ABSI score, TBSA, and the presence of third-degree burns—as the most critical predictors.

Crucially, our temporal validation analysis revealed that the model exhibited strong temporal stability, maintaining its high performance on a future patient cohort. However, we also identified a clinically significant decrease in sensitivity over time, highlighting that even robust models are subject to performance shifts. This central finding underscores the vital importance of implementing a robust governance framework, including continuous monitoring and periodic recalibration, to ensure that any predictive model remains safe and effective throughout its lifecycle in a clinical setting.

Future research should focus on validating these models on larger, multi-center datasets to enhance generalizability. Furthermore, exploring more advanced techniques, such as deep learning, could provide pathways to creating even more robust and adaptive predictive tools for burn care.

## Supporting information

**S1 Table. Features used for model development.** Detailed descriptive statistics and characteristics of all variables used in the study.
(DOCX)

**S2 Table. Hyperparameter tuning search space and optimal values.** Details of the grid search and selected parameters for each model.
(DOCX)

**S3 Table. Exploratory analysis of feature importance using multiple weighting criteria.** Ranking of features using multiple weighting criteria.
(DOCX)

**S4 Table. Confusion matrix of the final models.** Aggregated confusion matrices for GBT, RF, DT, DS, and RT models.
(DOCX)

**S5 Table. Full pairwise T-test p-values for model accuracy.** Statistical comparison of accuracy between models.
(DOCX)

**S6 Table. Full pairwise T-test p-values for the AUC metric.** Statistical comparison of AUC between models.
(DOCX)

**S7 Table. Full pairwise T-test p-values for the Brier score.** Statistical comparison of Brier scores between models.
(DOCX)

**S8 Table. Statistical comparison (p-values) of model performance with and without the ABSI feature.**

(DOCX)

**S1 Checklist. TRIPOD checklist.** Completed checklist demonstrating adherence to reporting guidelines for prediction model development and validation.
(DOCX)

## Acknowledgments

The authors would like to thank Isfahan University of Medical Sciences and all contributors to the burn inpatients registry project at the Injuries and Burn Subspecialized Teaching Hospital for their valuable support and collaboration.

## Author contributions

**Data curation:** Yasin Sabet Kouhanjani.

**Formal analysis:** Yasin Sabet Kouhanjani, Mohammad Sattari.

**Methodology:** Yasin Sabet Kouhanjani, Mohammad Sattari.

**Project administration:** Asghar Ehteshami.

**Supervision:** Asghar Ehteshami.

**Writing – original draft:** Yasin Sabet Kouhanjani.

**Writing – review & editing:** Yasin Sabet Kouhanjani, Mohammad Sattari, Asghar Ehteshami.

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
