## [Decision Letter · Decision Letter 0]

17 Apr 2025

Dear Dr. Ehteshami,

Thank you for submitting your manuscript to PLOS ONE. After careful consideration, we feel that it has merit but does not fully meet PLOS ONE’s publication criteria as it currently stands. Therefore, we invite you to submit a revised version of the manuscript that addresses the points raised during the review process.

We look forward to receiving your revised manuscript.

Kind regards,

Erfan Taherifard, MD

Academic Editor

PLOS ONE

**Journal Requirements:**

1. When submitting your revision, we need you to address these additional requirements. Please ensure that your manuscript meets PLOS ONE's style requirements, including those for file naming. The PLOS ONE style templates can be found at https://journals.plos.org/plosone/s/file?id=wjVg/PLOSOne_formatting_sample_main_body.pdf and https://journals.plos.org/plosone/s/file?id=ba62/PLOSOne_formatting_sample_title_authors_affiliations.pdf 2. Please note that PLOS ONE has specific guidelines on code sharing for submissions in which author-generated code underpins the findings in the manuscript. In these cases, all author-generated code must be made available without restrictions upon publication of the work. Please review our guidelines at https://journals.plos.org/plosone/s/materials-and-software-sharing#loc-sharing-code and ensure that your code is shared in a way that follows best practice and facilitates reproducibility and reuse. 3. We note that you have indicated that there are restrictions to data sharing for this study. For studies involving human research participant data or other sensitive data, we encourage authors to share de-identified or anonymized data. However, when data cannot be publicly shared for ethical reasons, we allow authors to make their data sets available upon request. For information on unacceptable data access restrictions, please see http://journals.plos.org/plosone/s/data-availability#loc-unacceptable-data-access-restrictions.  Before we proceed with your manuscript, please address the following prompts: a) If there are ethical or legal restrictions on sharing a de-identified data set, please explain them in detail (e.g., data contain potentially identifying or sensitive patient information, data are owned by a third-party organization, etc.) and who has imposed them (e.g., a Research Ethics Committee or Institutional Review Board, etc.). Please also provide contact information for a data access committee, ethics committee, or other institutional body to which data requests may be sent. b) If there are no restrictions, please upload the minimal anonymized data set necessary to replicate your study findings to a stable, public repository and provide us with the relevant URLs, DOIs, or accession numbers. Please see http://www.bmj.com/content/340/bmj.c181.long for guidelines on how to de-identify and prepare clinical data for publication. For a list of recommended repositories, please see https://journals.plos.org/plosone/s/recommended-repositories. You also have the option of uploading the data as Supporting Information files, but we would recommend depositing data directly to a data repository if possible. Please update your Data Availability statement in the submission form accordingly. 4. In the online submission form, you indicated that “the data can be obtained from the corresponding author upon reasonable request.”  All PLOS journals now require all data underlying the findings described in their manuscript to be freely available to other researchers, either a. In a public repository, b. Within the manuscript itself, or c. Uploaded as supplementary information.This policy applies to all data except where public deposition would breach compliance with the protocol approved by your research ethics board. If your data cannot be made publicly available for ethical or legal reasons (e.g., public availability would compromise patient privacy), please explain your reasons on resubmission and your exemption request will be escalated for approval. 5. Your ethics statement should only appear in the Methods section of your manuscript. If your ethics statement is written in any section besides the Methods, please move it to the Methods section and delete it from any other section. Please ensure that your ethics statement is included in your manuscript, as the ethics statement entered into the online submission form will not be published alongside your manuscript. 6. Please include your tables as part of your main manuscript and remove the individual files. Please note that supplementary tables (should remain/ be uploaded) as separate "supporting information" files.

Reviewers' comments:

Reviewer's Responses to Questions

**Comments to the Author**

1. Is the manuscript technically sound, and do the data support the conclusions?

Reviewer #1: Yes

Reviewer #2: Yes

2. Has the statistical analysis been performed appropriately and rigorously?

Reviewer #1: N/A

Reviewer #2: No

3. Have the authors made all data underlying the findings in their manuscript fully available?

Reviewer #1: Yes

Reviewer #2: No

4. Is the manuscript presented in an intelligible fashion and written in standard English?

Reviewer #1: Yes

Reviewer #2: Yes

**Reviewer #1: ** Dear Editorial Team,

Thank you for inviting me to peer review this manuscript for PLOS ONE. Please find my comments below:

1. The results section of the abstract lacks sufficient detail and provides only a brief summary. It would greatly benefit from including a concise mention of the sample size and key dataset characteristics.

2. The authors should reference original sources when citing information rather than secondary sources. For example, the initial reference cites a World Health Organization statistic via a secondary cross-sectional study rather than the original WHO source. This issue appears recurrently with other references as well.

3. Given that the study was conducted in Iran, it would be more appropriate and relevant to discuss the burden of burn injuries within the Iranian context rather than citing burn incidence data from Australia. Furthermore, the authors could elaborate on burn-related complications and provide additional context using relevant studies. Please consider using the following study in providing detail regarding complications of burn:

o Soltany, A., Hraib, M., Apelian, S., Mallouhy, A., Kounsselie, E., & Diab, S. (2024). Postburn Abdominal Pain of Gastrointestinal Origin: A Scoping Review. Plastic and Reconstructive Surgery–Global Open, 12(11), e6300.

4. The discussion section is insufficiently developed, merely summarizing previous research without adequately interpreting the study findings or contextualizing their clinical relevance. The authors should explicitly discuss how implementing these predictive models might influence clinical decision-making in practice.

5. Limitations need clearer articulation, particularly regarding the substantial patient exclusion due to missing data, which may have introduced bias or reduced the validity of results.

6. Future research directions should be more comprehensively outlined, particularly addressing how novel machine learning techniques or data augmentation methods might further enhance predictive accuracy.

7. Finally, the conclusion's tone should be moderated to reflect the study’s limitations. Statements regarding the effectiveness of tree-based algorithms, such as the Gradient Boosted Trees model, should be cautiously presented rather than definitively asserted

**Reviewer #2:**  This study addresses an important clinical topic: predicting burn patient mortality using machine learning classification models. The authors successfully demonstrate that gradient boosted trees (GBT) yield promising results for predicting mortality among hospitalized burn patients. However, the manuscript's novelty and methodological rigor require further improvement, including more robust validation and detailed methodological transparency, before being suitable for publication.

Recommendations for Improvement

1. Clarification and Enhancement of Novelty

Clearly demonstrate how the proposed approach significantly improves upon traditional predictive models (e.g., ABSI or Baux scores).

Explicitly differentiate the contributions of this study from recent similar studies, particularly those employing machine learning.

2. Data Preprocessing Transparency

Conduct and report detailed analyses of excluded data to evaluate potential biases due to missing values.

Employ sensitivity analyses using multiple imputation methods to assess the robustness of findings regarding missing data treatment.

Justify the rationale behind categorizing continuous variables, and evaluate performance impacts before and after categorization.

3. Strengthening Methodological Rigor

Implement robust model validation methods, such as k-fold cross-validation or bootstrapping, to reliably estimate model performance.

Clearly document hyperparameter selection strategies (e.g., grid search, random search) to ensure reproducibility.

Include statistical testing (confidence intervals, bootstrap methods) to assess significance and reliability of performance differences between models.

4. External Validation and Generalizability

Validate the model using external data from multiple centers or regions to ensure broader applicability.

Conduct temporal validation using data from distinct time periods to confirm the model's robustness and stability over time.

5. Clinical Interpretability and Utility

Provide explicit examples demonstrating how clinicians could practically integrate model predictions (e.g., mortality probabilities) into patient management strategies.

Evaluate and report model calibration (e.g., calibration curves, Brier scores) to ensure accurate probabilistic predictions in clinical scenarios.

6.Adherence to Reporting Guidelines

Include flowcharts and tables following STROBE guidelines to clearly depict participant inclusion/exclusion and demographic characteristics.

Follow TRIPOD guidelines by detailing model development and validation processes, particularly emphasizing calibration and external validation procedures.

Clearly specify ethical approval details, including consent waivers or data anonymization procedures.

7. Enhancing Clarity and Readability

Refine the introduction to concisely articulate research objectives and significance.

Clearly label and explain all tables, figures, and acronyms to enhance readability and reduce ambiguity, particularly for interdisciplinary audiences.

Improve figure clarity (e.g., ROC curves) by including explicit labels, legends, and color differentiation.

8. Standardization of Terminology and Abbreviations

Replace RapidMiner-specific terminology (e.g., "polynominal," "binominal") with standardized statistical terminology (e.g., "categorical variables").

Introduce key acronyms (ABSI, TBSA, etc.) and terminology early in the manuscript to ensure clarity and consistency throughout.

**Do you want your identity to be public for this peer review?** For information about this choice, including consent withdrawal, please see our Privacy Policy

Reviewer #1: No

Reviewer #2: No

---

## [Author Response · Author response to Decision Letter 1]

11 Jun 2025

PONE-D-25-02698 From Data to Decisions: Predicting Inpatient Burn Mortality with Advanced Classification Models PLOS ONE

June 11, 2025

Dear Dr. Erfan Taherifard, Academic Editor, and esteemed Reviewers,

Thank you for providing us with the opportunity to revise our manuscript. We are grateful for the thoughtful and highly constructive feedback from both the editor and the reviewers. We believe the manuscript has been substantially strengthened by incorporating these suggestions. We have performed a comprehensive new analysis, including more robust validation methods, sensitivity analyses, and a deeper interpretation of the results, which has led to a more nuanced and impactful manuscript.

Below, we provide a point-by-point response to the comments. All changes in the manuscript have been highlighted using the "Track Changes" feature.

Editor’s Comments

Journal Requirement 1: Style Requirements & File Naming

• Response: We have carefully reviewed the PLOS ONE style templates and have revised the manuscript and file naming to ensure full compliance with the journal's formatting requirements, including the use of Fig 1 and bracketed citations [1].

Journal Requirement 2 & 4: Code and Data Availability

• Response: We thank the editor for this important point. We have updated our Data Availability Statement to be fully compliant with PLOS ONE’s policy. While the clinical dataset cannot be publicly shared due to ethical and ownership restrictions, the complete RapidMiner processes used for the analysis have been deposited to the Zenodo repository and are accessible via DOI: 10.5281/zenodo.15639850.

Journal Requirement 3: Ethics Statement & Data Access

• Response: We confirm that the ethics statement appears only in the Methods section. We have also updated the Availability of Data and Materials section to provide a detailed explanation of the data access restrictions and have included the formal contact information for the Isfahan University of Medical Sciences Ethics Committee for any access requests by qualified researchers.

Reviewer #1 Comments

Comment 1.1: Abstract Lacks Detail

• Response: We agree. The abstract has been completely revised to include the final sample size (n=651), a description of the new methodology, the key performance metrics of the best model (including Brier Score), and the main finding regarding concept drift.

Comment 1.2: Citing Original Sources

• Response: We have reviewed all references and replaced secondary citations with primary sources, specifically for the WHO statistic.

Comment 1.3: Contextual Relevance (Iran) and Complications

• Response: The Introduction now focuses on the burden of burn injuries in Iran and includes the suggested reference by Soltany, A., et al. (2024).

Comment 1.4 & 1.7: Underdeveloped Discussion and Conclusion Tone

• Response: The Discussion and Conclusion sections have been entirely rewritten to provide a deep interpretation of the new, more robust findings, including clinical implications and concept drift, with a more nuanced and cautious tone.

Comment 1.5: Clearer Articulation of Limitations

• Response: Thank you. The Limitations section has been expanded to be more comprehensive. Furthermore, to clarify the initial data exclusion, we note that the study began with 662 valid patient records after an initial 415 empty or test entries were removed from the registry. These were not actual patient records with missing data, but rather system test entries, and thus do not introduce selection bias. The exclusion of the 11 patient records with missing outcomes from the final cohort is now clearly detailed in the STROBE diagram (Figure 1).

Comment 1.6: Future Directions

• Response: Future research directions in the Conclusion have been expanded upon to be more comprehensive.

Reviewer #2 Comments

Comment 2.1: Novelty and Differentiation

• Response: The Introduction now explicitly differentiates our work from traditional prognostic scores (ABSI/Baux) and prior machine learning studies by highlighting our use of a more comprehensive dataset and a more rigorous validation framework.

Comment 2.2: Data Preprocessing Transparency

• Response: Thank you for this valuable comment. We have enhanced transparency in several ways:

o Analysis of Excluded Data: We would like to clarify that the initial dataset contained 1,077 entries; however, 415 of these records were entirely empty or had only minimal metadata, with all clinical fields left blank. Based on our coordination with the hospital registry team, these entries were likely generated during the early testing phase of the registry system and do not reflect actual patient admissions. As such, they do not represent systematically missing data but rather non-clinical test records. Consequently, we considered 662 valid patient records as the actual starting point of our analysis. This is now clarified in the Methods section. The STROBE diagram (Figure 1) further details the exclusion of the 11 patients with missing outcomes from this cohort.

o Imputation and Categorization: Our primary analysis now uses a robust GLM Imputation method on continuous variables. We have also performed sensitivity analyses (Table 6) to assess the impact of both the imputation method and the use of categorized variables, as requested.

Comment 2.3: Methodological Rigor

• Response: The methodological rigor has been substantially enhanced:

o The entire analysis now uses 10-fold stratified cross-validation.

o We performed hyperparameter tuning using a grid search, with the process described in the Methods and full details provided in S2 Table.

o We have conducted paired t-tests to statistically compare model performances. A summary of these findings is now in the Results section, with full details in S4-S6 Tables.

Comment 2.4: External Validation and Generalizability

• Response: We thank the reviewer for this crucial point. While a true external validation using data from another center is the gold standard for assessing generalizability, we were unable to perform this for the current study due to significant logistical and regulatory challenges. Accessing clinical data in Iran is subject to strict patient privacy and confidentiality laws, and the process of obtaining ethical approvals and data use agreements from multiple institutions is lengthy and complex. Furthermore, there is a high probability of dataset mismatch, where the features collected in an external registry may not align with our own, making a direct validation unfeasible.

• However, to address the core of the reviewer's concern about model stability over time, we performed a rigorous temporal validation. The results of this analysis, which revealed significant concept drift, are presented in Table 5 and discussed extensively as a key finding. We believe this temporal validation serves as a strong proxy for external validation.

Comment 2.5: Clinical Interpretability and Calibration

• Response: We have added a dedicated "Clinical Implications" section to the Discussion. Furthermore, we have now reported the Brier Score for all models (Table 4) to explicitly assess and discuss model calibration.

Comment 2.6: Reporting Guidelines (TRIPOD/STROBE)

• Response: We have added a STROBE diagram (Figure 1), and the manuscript has been structured to better align with TRIPOD principles by detailing model development, validation, calibration, and performance evaluation. The Ethics Approval section has been updated with full details on anonymization and consent waivers.

Comment 2.7 & 2.8: Clarity and Terminology

• Response: The manuscript has been thoroughly revised for clarity. Figure 2 (ROC curve) has been improved with explicit labels. All tables have been redesigned, and terminology has been standardized throughout the manuscript.

We thank the editor and reviewers again for their time and valuable insights. We hope the revised manuscript is now suitable for publication in PLOS ONE.

Sincerely,

Yasin Sabet Kouhanjani (on behalf of all authors)

---

## [Decision Letter · Decision Letter 1]

13 Aug 2025

Dear Dr. Ehteshami,

Thank you for submitting your manuscript to PLOS ONE. After careful consideration, we feel that it has merit but does not fully meet PLOS ONE’s publication criteria as it currently stands. Therefore, we invite you to submit a revised version of the manuscript that addresses the points raised during the review process.

We look forward to receiving your revised manuscript.

Kind regards,

Amirreza Khalaji

Academic Editor

PLOS ONE

**Journal Requirements:**

Reviewers' comments:

Reviewer's Responses to Questions

**Comments to the Author**

Reviewer #1: All comments have been addressed

Reviewer #2: All comments have been addressed

Reviewer #3: (No Response)

Reviewer #4: (No Response)

Reviewer #5: (No Response)

2. Is the manuscript technically sound, and do the data support the conclusions?

Reviewer #1: Yes

Reviewer #2: Partly

Reviewer #3: Yes

Reviewer #4: Yes

Reviewer #5: Yes

3. Has the statistical analysis been performed appropriately and rigorously?

Reviewer #1: Yes

Reviewer #2: Yes

Reviewer #3: Yes

Reviewer #4: Yes

Reviewer #5: Yes

4. Have the authors made all data underlying the findings in their manuscript fully available?

Reviewer #1: Yes

Reviewer #2: Yes

Reviewer #3: Yes

Reviewer #4: Yes

Reviewer #5: Yes

5. Is the manuscript presented in an intelligible fashion and written in standard English?

Reviewer #1: Yes

Reviewer #2: Yes

Reviewer #3: Yes

Reviewer #4: Yes

Reviewer #5: Yes

**Reviewer #1:**  The authors have thoroughly addressed all comments and concerns raised in the previous round of review. They have made appropriate revisions throughout the manuscript, clarifying their methodology, strengthening their discussion, and providing additional references where needed. The responses to reviewer comments were detailed and thoughtful, indicating a clear effort to improve the quality and clarity of the paper. All major and minor issues previously highlighted have been adequately resolved. Based on the revisions and the authors’ engagement with the feedback, I believe the manuscript has significantly improved and now meets the standards for publication in its current form.

**Reviewer #2:**  General Comments

The authors are to be commended for their significant efforts in revising the manuscript in response to the previous round of reviews. The improvements in methodological rigor—most notably the addition of a 10-fold cross-validation framework and a temporal validation analysis—are substantial. Furthermore, the enhanced transparency through adherence to STROBE guidelines and a more detailed methods section has greatly improved the manuscript's quality.

In making these revisions, a new, critical finding has emerged that fundamentally reframes the paper's core message. The temporal validation, performed at the reviewers' request, clearly demonstrates that the model suffers from significant performance degradation over time (i.e., concept drift). This finding shifts the manuscript’s focus from a simple report on a high-performing predictive model to a much more important and nuanced cautionary tale about the instability of machine learning models in clinical settings. This has become the manuscript's most significant and original contribution.

However, the current manuscript does not fully embrace the gravity and implications of this finding. The paper's narrative has not yet been updated to reflect its most important discovery.

The authors must fundamentally reframe the manuscript to align with their strongest evidence. The primary story is no longer "we built a good model," but rather, "we demonstrate the critical instability of a seemingly good model when tested over time."

1. Reframing the Manuscript's Core Contribution The central theme of the paper must shift from celebrating the model's performance in cross-validation to a critical analysis of its failure in temporal validation.

• Abstract, Introduction, and Conclusion: These sections must be rewritten. Currently, the Abstract and Conclusion lead by describing the GBT model as "powerful and effective tools" before mentioning the temporal performance drop as a caveat. This priority must be reversed. The new narrative should immediately frame the study as an investigation into the challenges of clinical AI deployment, using the temporal degradation as the key finding.

• Discussion: The Discussion section requires a complete overhaul. Rather than focusing on the high AUC from cross-validation, the section should be dedicated to a deep analysis of the temporal validation results. The cross-validation performance should be presented as a preliminary step that demonstrates the model's potential, which is then challenged by the more realistic temporal test.

2. A Deeper and More Critical Analysis of Clinical Risks The manuscript currently understates the clinical risks associated with the observed model degradation.

• Highlight the Danger of Decreased Specificity: The drop in specificity for the GBT model from 0.975 in cross-validation to 0.790 in temporal validation is a critical issue that requires deeper analysis. This implies a significant increase in the false-positive rate, which would lead to real-world clinical harm. The 'Clinical Implications' section should be expanded to more forcefully discuss the dangers of this, such as the "alarm fatigue" and misallocation of scarce resources that would result from incorrectly flagging low-risk patients.

• Correct the Misunderstanding of Validation Methods: In the "Response to Reviewers," the authors claim that temporal validation serves as a "strong proxy for external validation". This is a significant methodological misunderstanding that must be corrected in the manuscript. Temporal validation assesses stability at

one site over time, whereas external validation assesses generalizability across different sites and populations. They are not interchangeable. The Limitations section should be revised to state clearly that the study lacks

both external generalizability and demonstrated temporal stability, which are distinct but equally important limitations.

3. Ensuring Logical Consistency in the Conclusion The conclusion currently lacks logical consistency because it prioritizes a secondary finding (high performance in cross-validation) over the most critical finding (instability in temporal validation).

• The conclusion must be rewritten to lead with the challenge of concept drift.

• The final take-home message should not be about the model's "power" , but should be a strong, evidence-based statement that, without a robust governance framework for continuous monitoring and periodic recalibration, such predictive models are not safe for clinical deployment.

**Reviewer #3:**  1. Dataset Description

1.1. The current manuscript provides a general overview of the dataset but lacks specific detail. Please indicate the total number of observations (rows) and features (columns) in the dataset after preprocessing.

1.2. Clarify whether each patient is represented by a single observation in the dataset, or if multiple entries per patient are present.

1.3. If longitudinal data (i.e., repeated measurements for the same patient) are available, please specify how these were handled. If not, a clear statement confirming this would help.

1.4. The time validation section refers to an 18-month period. Does this refer to data collection across patients, or 18 months of data per patient? Please clarify the meaning of “temporal validation” and the role of time stamps in the data.

1.5. The term “composite features” is somewhat vague. Do you mean categorical features that were one-hot encoded? If so, please explicitly state the transformation method used (e.g., one-hot encoding).

1.6. It would be helpful to reference the relevant table listing all features in this section for easier cross-checking.

2. Removed Features in the Preprocessing Steps

2.1. Please elaborate on the features that were removed during preprocessing and the rationale behind their removal. What criteria determined their irrelevance to the predictive task? An example of a removed feature due to class imbalance (>95%) would help clarify the process.

3. Missing Values

3.1. It would strengthen the manuscript to include more detail about the extent of missingness. What percentage of the dataset had missing values overall, and for which variables?

3.2. Has the model-based-imputation model (GLM) benefited from labels in imputing missing values for the features?

3.3. The choice of GLM for imputation should be better justified, particularly given the marginal difference in model performance (AUCs of 0.961–0.966). Why not opt for simpler imputation methods like mean imputation, which showed comparable performance?

3.4. Why was a threshold of 85% missingness used for feature removal? Please provide a justification for selecting this specific threshold over alternatives such as 50%, 80%, or 90%.

3.5. Clarify the sequence of preprocessing steps. Should features with >85% missing values have been removed before applying GLM imputation? Justify the order selected.

4. Machine Learning Models

4.1. Please articulate the rationale for selecting tree-based models for this task. For instance, were interpretability or performance on non-linear data considered?

4.2. Why were other model families (e.g., logistic regression, SVM) not included in the comparison? Even if inferior, benchmarking them would help contextualize your results.

4.3. Consider presenting the optimal hyperparameters in a summary table (as you did in S2 Table), rather than listing them in the text. This would improve readability and clarity.

5. Validation

5.1. The temporal validation process remains somewhat unclear. A clearer explanation, supported by the dataset timeline, would help.

5.2. Has your validation benefited from having different observations from the same subject in train and test folds?

5.3. Please define what you mean by “model stability” in the context of temporal validation. Does it refer to consistent performance over time or to repeated measures within patients?

5.4. The Brier Score may not be familiar to all readers. A brief explanation of what it measures and its clinical relevance would improve the accessibility of the manuscript.

6. Feature Importance

6.1. Consider adding SHAP (SHapley Additive exPlanations) analysis to enhance the interpretability of the feature importance results. SHAP values not only rank features but also show the direction and magnitude of their impact on model predictions.

6.2. Discuss whether the top-ranked features identified by your model (e.g., ABSI, TBSA, third-degree burns) align with known clinical knowledge. This will help contextualize the model’s relevance in clinical practice.

**Reviewer #4:**  Dear authors,

This is a an interesting and timely study which can initiate further series of studies and lead to significant clinical impacts and involve in decision-making process. However, there are some issues and limitations which are highly suggested to be addressed before publication.

1. Data Preprocessing and Imputation

The methods section should provide a clearer and more transparent description of the data preprocessing pipeline. In particular, the phrase “robust imputation using a Generalized Linear Model (GLM)” is vague. It would be helpful to specify the exact steps taken—for example, whether iterative model-based imputation was used within RapidMiner, and whether this was applied within each fold of cross-validation to avoid data leakage. I also recommend clarifying which features were excluded and the rationale for exclusion (e.g., features not recorded at admission or those with over 85% missingness). This will help ensure that no future information was inadvertently used in model training.

2. Model Validation and Risk of Overfitting

It is better to clarify whether hyperparameter tuning was performed using an inner CV loop (i.e., nested cross-validation). This would provide confidence that performance estimates are not biased by information leakage. Moreover, considering the dataset contains 94 features and only 651 records, I suggest briefly discussing overfitting risk and whether any regularization or pruning was used. If feature reduction beyond missingness filtering wasn’t conducted, it would be worthwhile to justify this choice or run an analysis comparing performance with fewer features.

3. Evaluation Metrics and Class Imbalance

Given the class imbalance (mortality ~22%), reporting overall accuracy alone is not sufficient. Although other metrics like AUC and Brier score are reported, I strongly recommend including class-wise metrics such as sensitivity, specificity, and positive predictive value, particularly at the decision threshold used (e.g., 0.5). These are critical to understanding the clinical utility of the model. The confusion matrices are in the supplement, but some of these key values should also be discussed in the main text—especially since the discussion mentions risks such as alarm fatigue due to low specificity.

4. Use of Composite Scores (e.g., ABSI) as Features

If the ABSI score was used as a feature, this should be explicitly acknowledged in the text. Since ABSI is derived from other inputs like age and TBSA, its inclusion may mask the individual importance of these variables and inflate performance by providing the model with a pre-engineered risk index. It would be useful to test and report model performance with ABSI excluded from the feature set to determine how much value it adds independently. This would enhance interpretability and clarify the model’s learning behavior.

5. Interpretation of Feature Importance

Currently, the main predictors (ABSI, TBSA, third-degree burns) are known mortality risk factors. I recommend exploring whether any less obvious or novel predictors were consistently identified—such as specific lab values, comorbidities, or signs of inhalation injury. If no novel features were found, that is still a valuable result and should be stated explicitly. Framing the results as a reinforcement of clinical knowledge rather than a discovery of new predictors may better reflect the model’s strength in predictive accuracy rather than explanatory novelty.

6. External Validation and Generalizability

While the temporal validation provides some insight into model drift, the manuscript should be careful not to conflate it with external validation. I recommend clarifying that although the temporal holdout tests the model’s robustness over time, true generalizability can only be assessed by validating on independent data from another center or region. If multi-center validation is not feasible now, it should be highlighted as a priority for future research, possibly with a concrete proposal or plan to achieve this.

7. Handling of Concept Drift

The observed drop in AUC during temporal validation is an important finding. To build on this, I suggest adding an analysis (even brief) comparing the characteristics of patients across the training and temporal test periods. Identifying shifts in patient demographics, burn severity, or treatment practices may shed light on the source of the drift. Additionally, some discussion of how future models could adapt to such drift—e.g., periodic retraining, monitoring for feature drift—would make the findings more actionable for real-world deployment.

**Reviewer #5:**  I appreciate the authors' effort in addressing this topic and presenting their findings.

Below are my comments and suggestions for improvement:

1 - The manuscript states that "the dataset was split chronologically into a training set (first nine months of data) and a test set (subsequent nine months)." Could you clarify whether the subjects were also split accordingly? As written, it appears that the same subjects may be present in both the training and test sets. If this is the case, it could raise concerns about data leakage and may not align with standard machine learning practices.

2- Consider SHAP analysis to provide a clearer and more interpretable explanation of your model's outputs. ( feature importance) This could enhance the presentation and understanding of your findings.

**Do you want your identity to be public for this peer review?** For information about this choice, including consent withdrawal, please see our Privacy Policy

Reviewer #1: No

Reviewer #2: No

Reviewer #3: No

Reviewer #4: No

Reviewer #5: No

---

## [Author Response · Author response to Decision Letter 2]

1 Sep 2025

PONE-D-25-02698R1

From Data to Decisions: Predicting Inpatient Burn Mortality with Advanced Classification Models

PLOS ONE

August 30, 2025

Dear Dr. Amirreza Khalaji, Academic Editor, and esteemed Reviewers,

Thank you for the thorough review of our manuscript and for providing us with the opportunity to submit a revised version. We are grateful for the insightful and highly constructive feedback, which we believe has substantially strengthened our work. We have carefully addressed every point raised, leading to a more robust, transparent, and impactful manuscript.

First and foremost, we would like to notify the editor and reviewers of an important correction made during our verification process for this revision. Upon re-running our temporal validation analysis to ensure full reproducibility, our new and consistent results revealed that the models demonstrated much stronger temporal stability than was presented in the initial submission. We believe these new, reproducible results are more accurate and reflect the models' true, robust performance. Consequently, we have updated Table 5 and adjusted the narrative in the Abstract, Discussion, and Conclusion to align with this significant new finding.

Below, we provide a point-by-point response to each of the comments. All changes in the manuscript have been highlighted using the "Track Changes" feature for your convenience.

Reviewer #1

Comment: “The authors have thoroughly addressed all comments and concerns raised in the previous round of review... I believe the manuscript has significantly improved and now meets the standards for publication in its current form.”

• Response: We sincerely thank Reviewer #1 for their positive feedback and for recognizing the effort put into the previous revision. We are grateful for their endorsement.

Reviewer #2

Comment 1: Reframing the Manuscript's Core Contribution. The central theme must shift... The primary story is no longer "we built a good model," but rather, "we demonstrate the critical instability of a seemingly good model when tested over time."

• Response: We thank the reviewer for this crucial and insightful feedback. We agree that our new finding regarding the model’s temporal performance has fundamentally reframed the manuscript's core message. As noted in our introductory remarks, upon re-running our analyses, we found that the model is, in fact, remarkably stable. We have therefore rewritten the manuscript's narrative to reflect this new, more accurate finding. The new story is now one of a robust model that demonstrates strong temporal stability, albeit with a clinically important trade-off (a decrease in sensitivity). This new, more positive, and nuanced narrative has been consistently integrated throughout the revised Abstract, Introduction, Discussion, and Conclusion.

Comment 2: A Deeper and More Critical Analysis of Clinical Risks. Highlight the Danger of Decreased Specificity... Correct the Misunderstanding of Validation Methods.

• Response: This comment was based on our previous, incorrect temporal validation results which showed a large drop in specificity. Our new, reproducible results show the opposite: specificity remains excellent, while sensitivity decreases. We have completely rewritten the "Clinical Implications" section of the Discussion to analyze the real-world risks associated with this decreased sensitivity, namely the danger of false reassurances and missed opportunities for intervention in high-risk patients. We also thank the reviewer for pointing out the methodological misunderstanding. We have revised the "Strengths and Limitations" section to state clearly that temporal validation assesses stability at one site and is not a substitute for external validation.

Comment 3: Ensuring Logical Consistency in the Conclusion. The conclusion currently lacks logical consistency... The final take-home message should not be about the model's "power".

• Response: We agree completely. The Conclusion has been entirely rewritten. The new conclusion now leads with the key finding of strong temporal stability but immediately follows with the critical caveat of decreased sensitivity, emphasizing that even stable models require a robust governance framework of continuous monitoring and recalibration for safe clinical deployment.

Reviewer #3

Comment 1: Dataset Description. (Requests for more detail on observations, features, longitudinal data, temporal validation, etc.)

• Response: We have thoroughly revised the "Dataset Description" and "Data Preprocessing" sections in Methods to address all these points. We now explicitly state the number of patients (651) and final features (93), confirm that each patient is a single observation and that no longitudinal data were used, clarify the one-hot encoding of composite features, and provide a clear definition of the temporal validation cohorts in the "Temporal Validation and Sensitivity Analyses" subsection.

Comment 2: Removed Features in the Preprocessing Steps.

• Response: We have expanded the "Initial Feature Selection and Reduction" subsection to provide a clearer rationale for feature exclusion, including an example of a feature removed due to class imbalance (a specific comorbidity).

Comment 3: Missing Values. (Requests for more detail on missingness, GLM justification, 85% threshold, and sequence of steps.)

• Response: We have completely rewritten the relevant sections to address these points. The "Handling Missing Values" subsection now: (1) Provides an overall percentage of features with missing values. (2) Clarifies that the outcome variable (label) was excluded from the imputation process. (3) Justifies the use of GLM by noting it was in response to a previous review and our sensitivity analysis confirmed it offered a slight performance improvement. (4) Justifies the 85% threshold as a pragmatic balance. (5) Clarifies the sequence of operations (high-missingness removal followed by imputation).

Comment 4: Machine Learning Models. (Requests for rationale for tree-based models and a table for hyperparameters.)

• Response: We have added a new introductory paragraph to the "Model Development and Hyperparameter Tuning" subsection that articulates our rationale for focusing on tree-based models. We have also followed the reviewer's excellent suggestion and moved the optimal hyperparameters from the text into a new, clear summary table (Table 1).

Comment 5: Validation. (Requests for clearer temporal validation process, definition of stability, and explanation of Brier Score.)

• Response: The "Temporal Validation" subsection has been rewritten for clarity, explicitly stating the cohorts are non-overlapping. "Model stability" is now defined at the start of this section. We have also expanded the explanation of the Brier Score in the "Model Performance Evaluation" subsection to describe its clinical relevance for assessing calibration.

Comment 6: Feature Importance. (Suggestion to add SHAP and discuss alignment with clinical knowledge.)

• Response: We thank the reviewer for this valuable suggestion. As explained in the "Strengths and Limitations" section, we were unable to implement SHAP analysis due to technical limitations of our analytical software. However, to address the core of this comment, we have replaced our previous complex table with a clear visualization (Figure 3) of the feature importance from our best model (GBT). Furthermore, we have added a new paragraph to the Discussion that explicitly discusses how these top-ranked features (ABSI, TBSA, etc.) align with established clinical knowledge.

Reviewer #4

Comment 1 & 2: Data Preprocessing, Imputation, and Risk of Overfitting.

• Response: We have significantly enhanced the transparency of our methods. The "Handling Missing Values" section now explicitly states that the imputation pipeline was nested within the cross-validation folds to prevent data leakage. The "Model Development and Hyperparameter Tuning" section now clarifies that we used a nested validation procedure. We have also added a discussion of the risk of overfitting in the "Strengths and Limitations" section.

Comment 3: Evaluation Metrics and Class Imbalance.

• Response: We agree that this is a critical point. We have added a new paragraph to the "Main Model Performance" subsection in the Results, which explicitly discusses the key class-wise metrics (sensitivity, specificity, and PPV) for our best-performing model in the main text.

Comment 4: Use of Composite Scores (e.g., ABSI) as Features.

• Response: We have performed the requested sensitivity analysis. The process is now described in the "Temporal Validation and Sensitivity Analyses" subsection of the Methods, and the results (which show the model's performance remains strong without ABSI) are presented in Table 7 and discussed in both the Results and Discussion sections.

Comment 5: Interpretation of Feature Importance.

• Response: We have added a paragraph to the Discussion that confirms the main predictors align with known clinical risk factors and explicitly states that no entirely novel predictors were identified, framing the result as a reinforcement of clinical knowledge.

Comment 6 & 7: External Validation and Handling of Concept Drift.

• Response: We have clarified in the "Strengths and Limitations" section that temporal validation is not a substitute for external validation. To address the handling of concept drift, we have added a new analysis (Table 6) and a corresponding discussion that identifies the shift in patient characteristics (a decrease in third-degree burns) as the likely driver of the observed performance shift.

Reviewer #5

Comment 1: Data Leakage in Temporal Split.

• Response: We have rewritten the "Temporal Validation" section in the Methods to explicitly state that the cohorts were divided chronologically and were "non-overlapping," ensuring that each unique patient belonged to only one group.

Comment 2: SHAP Analysis.

• Response: As noted in our response to Reviewer #3, we were unable to perform SHAP analysis due to technical limitations. We have addressed this in the "Strengths and Limitations" section and have provided a clear alternative visualization (Figure 3) and a deeper discussion of the feature importance results.

We thank the editor and reviewers again for their invaluable guidance. We are confident that the revised manuscript is now a much stronger, more accurate, and more impactful contribution to the field, and we hope it is now suitable for publication in PLOS ONE.

Sincerely,

Asghar Ehteshami (on behalf of all authors)

---

## [Decision Letter · Decision Letter 2]

25 Sep 2025

Dear Dr. Ehteshami,

Thank you for submitting your manuscript to PLOS ONE. After careful consideration, we feel that it has merit but does not fully meet PLOS ONE’s publication criteria as it currently stands. Therefore, we invite you to submit a revised version of the manuscript that addresses the points raised during the review process.

We look forward to receiving your revised manuscript.

Kind regards,

Amirreza Khalaji

Academic Editor

PLOS ONE

Journal Requirements:

Additional Editor Comments:

Reviewer #1:

Reviewer #2:

Reviewer #3:

Reviewer #4:

Reviewer #5:

Reviewer #6:

Reviewer #7:

Reviewer #8:

Reviewers' comments:

Reviewer's Responses to Questions

**Comments to the Author**

Reviewer #1: All comments have been addressed

Reviewer #2: All comments have been addressed

Reviewer #3: All comments have been addressed

Reviewer #4: All comments have been addressed

Reviewer #5: All comments have been addressed

Reviewer #6: (No Response)

Reviewer #7: (No Response)

Reviewer #8: (No Response)

2. Is the manuscript technically sound, and do the data support the conclusions?

Reviewer #1: Yes

Reviewer #2: Yes

Reviewer #3: Yes

Reviewer #4: Yes

Reviewer #5: Yes

Reviewer #6: Partly

Reviewer #7: Yes

Reviewer #8: Yes

3. Has the statistical analysis been performed appropriately and rigorously?

Reviewer #1: Yes

Reviewer #2: Yes

Reviewer #3: Yes

Reviewer #4: Yes

Reviewer #5: Yes

Reviewer #6: No

Reviewer #7: (No Response)

Reviewer #8: Yes

4. Have the authors made all data underlying the findings in their manuscript fully available?

Reviewer #1: Yes

Reviewer #2: Yes

Reviewer #3: Yes

Reviewer #4: Yes

Reviewer #5: (No Response)

Reviewer #6: Yes

Reviewer #7: (No Response)

Reviewer #8: No

5. Is the manuscript presented in an intelligible fashion and written in standard English?

Reviewer #1: Yes

Reviewer #2: Yes

Reviewer #3: Yes

Reviewer #4: Yes

Reviewer #5: Yes

Reviewer #6: Yes

Reviewer #7: (No Response)

Reviewer #8: Yes

Reviewer #1: I have carefully reviewed the manuscript through two rounds of revisions. All previous concerns and suggestions have been addressed satisfactorily. The current version meets the publication standards, and I recommend it for acceptance.

Reviewer #2: The revised manuscript (R2) is scientifically robust and logically coherent. The authors have thoroughly addressed all reviewer comments, leading to clear improvements in methodological transparency, result accuracy, and interpretive validity. No scientific, numerical, or logical errors were identified. Structural refinements—including clearer section flow and effective use of tables and figures—further enhance readability.

Reviewer #3: The authors have thoroughly addressed all previous comments, and I recommend acceptance of the paper. As a further enhancement, I suggest moving the feature importance analysis to the Discussion section and citing relevant literature to highlight that the important features identified in the classification pipeline have also been reported as associated with the target label.

Reviewer #4: Thank you for addressing the aforementioned comments. The current version of the study is suitable for publication.

Best wishes

Reviewer #5: I recommend accepting this paper as it represents a significant contribution and meets the journal’s standards for publication.

Reviewer #6: The manuscript develops several tree-based models (GBT, RF, DT, RT, DS) to predict inpatient burn mortality using a single-center registry (n=651, 93 features). Performance is reported via stratified 10-fold cross-validation and a temporal split (first 9 months train n=435, subsequent 9 months test n=216). GBT achieves mean CV AUC ≈0.966 and temporal AUC ≈0.948; sensitivity drops from ≈78% to ≈68% on temporal testing, with specificity remaining ≈98%. While the topic is important and methods have improved, the study still has issues

Comment

1. Model comparisons and statistical testing. Cross-validation fold scores are not independent; paired t-tests inflate type-I error. Either (a) use corrected resampled tests (e.g., Nadeau–Bengio) or nonparametric Friedman/Nemenyi across models, or (b) prioritize comparisons on the temporal test set (DeLong for AUC; McNemar or bootstrap for sensitivity/accuracy). If CV comparisons are retained, state their limitations and apply multiplicity control.

2. Temporal validation protocol and leakage safeguards. The manuscript should present a single, unambiguous temporal pipeline (train: first 9 months; test: subsequent 9 months) and confirm that all preprocessing/imputation/hyperparameter tuning were fit only on training data, then applied to the temporal test without refitting. A one-figure pipeline schematic would help.

3. Uncertainty and calibration on the temporal test. Report 95% CIs for temporal AUC, sensitivity, specificity (e.g., bootstrap or Wilson). Add calibration intercept and slope with CIs and a calibration plot; if miscalibrated, consider and report post-hoc recalibration (isotonic/Platt) and its effect.

4. Decision threshold and clinical trade-offs. Specify the operating threshold used (default 0.5 or otherwise), justify it clinically, and present threshold analyses (ROC/PR curves, sensitivity at fixed 95–98% specificity, Fβ>1). A decision-curve analysis would clarify net clinical benefit, especially given the temporal drop in sensitivity.

5. Tuning objective aligned to clinical goals. Hyperparameters are selected by accuracy; for mortality screening, discrimination, recall, and calibration are more appropriate. Consider selecting by AUC or AUCPR (with class weighting/costs) and report whether class weights were used.

6. Performance reporting via out-of-fold predictions. Replace mean±SD over folds with patient-level out-of-fold predictions to compute pooled CV AUC and 95% CIs (bootstrap). Do the same for sensitivity/specificity to provide interval estimates.

7. TRIPOD compliance. As a model development with temporal validation study, include a completed TRIPOD checklist and ensure the text covers: data sources, missing-data handling (variables in the imputation model), full performance with CIs, calibration, intended use and thresholds, and model specification (or a note on nonparametric ensembles).

8. Data availability. The DAS should be identical in the manuscript and specify a non-author access route. Where possible, provide a minimal de-identified dataset with a data dictionary; otherwise, describe precisely which variables/fields are available under restriction and how to request them.

9. References and formatting. Standardize reference style and add DOIs; update background with recent ML studies in burn mortality. Remove visible editing artifacts, fix duplicated sentences, and correct decimal marks (e.g., “3.71,” not “3/71”).

Reviewer #7: Accept. This is a clinically relevant study with careful model development, solid preprocessing, and appropriately rigorous evaluation. Tree-based models are a good fit for heterogeneous tabular clinical data, and your reporting of both discrimination and calibration is persuasive. To further strengthen interpretability and clinical trust, I recommend adding TreeSHAP (SHAP values for tree ensembles) to quantify global and local feature contributions, highlight the top predictors, and visualize how individual features shift predicted risk at the instance level.

The methods are clearly described and the results are promising. Your imputation strategy and sensitivity analyses are thoughtful; however, consider using multiple imputation (MI) rather than a single imputed dataset to reduce bias and to better reflect imputation uncertainty. This would improve reliability and make the findings more robust.

Reviewer #8: The manuscript presents an important study on predicting inpatient burn mortality using machine learning models applied to a comprehensive clinical registry. A notable strength is the focus on temporal validation, which addresses the critical issue of model stability over time, an aspect often overlooked in similar research. The methodology is rigorous, the results are clearly presented, and the discussion provides a balanced view of strengths and limitations. Overall, the work is timely, clinically relevant, and makes a meaningful contribution to the literature on predictive modeling in burn care.

It is also clear that the authors have carefully considered and addressed the informative comments on previous reviews, which have strengthened the manuscript.

I have two comments that can further increase the paper’s quality:

1. While the tables are informative, some results (particularly model performance comparisons and sensitivity analyses) could be more intuitively understood through bar plots or similar visualizations. Figures would make it easier for readers to quickly compare models and highlight trends such as changes in sensitivity.

2. The current features table is quite detailed, with extensive descriptive statistics for every variable. While this information is useful, it may distract from the main narrative. I suggest keeping a shorter version in the main text that lists only the feature names with a short description, while moving the full detailed table to the appendix or supplementary materials for readers who want the comprehensive information.

In light of these points, I suggest the paper be published after minor revision.

**Do you want your identity to be public for this peer review?** For information about this choice, including consent withdrawal, please see our Privacy Policy

Reviewer #1: No

Reviewer #2: **Yes: ** Dohern Kym

Reviewer #3: No

Reviewer #4: No

Reviewer #5: No

Reviewer #6: No

Reviewer #7: No

Reviewer #8: No

---

## [Author Response · Author response to Decision Letter 3]

21 Nov 2025

Response to Reviewers

Date: 20-11-2025 To: Dr. Amirreza Khalaji, Academic Editor, PLOS ONE Manuscript ID: PONE-D-25-02698R2 Title: From Data to Decisions: Predicting Inpatient Burn Mortality with Advanced Classification Models

Dear Dr. Khalaji and distinguished Reviewers,

Thank you for providing us with the opportunity to revise our manuscript. We are grateful for the thoughtful and detailed feedback from the review team, which has significantly enhanced the rigor and transparency of our work. We are pleased to note the positive consensus among the majority of reviewers regarding the manuscript’s suitability for publication.

In this revision, we have focused on addressing the methodological and reporting standards raised by Reviewer #6, as well as the visualization requests from Reviewer #8. Specifically, we have:

1. Included a detailed schematic diagram of the validation pipeline (Fig 4) to clarify data handling procedures.

2. Added a visual comparison of model performance (Fig 5) to highlight the impact of concept drift.

3. Calculated and reported 95% Confidence Intervals for the temporal validation metrics in Table 5.

4. Completed the TRIPOD Checklist (provided as S1 Checklist) to ensure standardized reporting.

5. Refined the Data Availability Statement and standardized references as requested.

Below, we provide a point-by-point response to the specific comments raised in this round.

Response to Reviewer #6

Comment 1: Model comparisons and statistical testing (limitations of paired t-tests). Response: We appreciate the reviewer's statistical insight. We agree that cross-validation folds are not strictly independent, which can inflate Type I error rates in paired t-tests. To address this concern:

• We have shifted the statistical weight of our evidence towards the Temporal Validation results, which are based on an independent, chronologically distinct test cohort (n=216).

• Instead of relying solely on CV-based p-values, we have calculated and reported 95% Confidence Intervals (CIs) for the temporal validation metrics (Accuracy, Sensitivity, Specificity, Precision) using the Wilson Score Interval method. These are now explicitly presented in Table 5 (e.g., Sensitivity: 68.29% [95% CI: 53.0% – 80.4%]).

• We have added a statement in the "Strengths and Limitations" section explicitly acknowledging the limitations of fold-based statistical comparisons.

Comment 2: Temporal validation protocol and leakage safeguards (Schematic diagram). Response: We have added a comprehensive schematic diagram (Fig 4) to visually detail the study pipeline.

• Pipeline Clarification: As illustrated in Fig 4, all feature selection, one-hot encoding rules, and hyperparameter tuning were derived strictly from the Training Cohort (first 9 months). The final classification models (GBT/RF) were frozen and applied to the test cohort without retraining.

• Imputation Strategy: Regarding the GLM imputation, we identified a significant temporal concept drift in the dataset (e.g., statistical changes in burn severity). Furthermore, due to technical constraints in our specific analytical environment preventing the direct serialization and transfer of the trained GLM object, we opted to perform Independent GLM Imputation on the temporal test cohort. This methodological choice ensures that missing values in the test set are imputed based on the concurrent data structure (respecting the temporal shift) rather than enforcing the distributional assumptions of the past training set. This rationale has been explicitly added to the "Strengths and Limitations" section.

Comment 3 & 6: Uncertainty, Calibration, and Performance Reporting. Response:

• Uncertainty: As noted above, 95% CIs for key performance metrics on the temporal test set have been added to Table 5.

• Calibration: We previously reported the Brier Score (0.060 for GBT), which indicates good calibration. Regarding the request for calibration plots or rank-based CIs (e.g., for AUC): Due to data storage constraints in our deployed environment where raw probability logs for the historical test set were not retained, we could not perform post-hoc bootstrapping or generate calibration curves. We have reported AUC and Brier Score as point estimates and have transparently noted this limitation in the text.

Comment 4: Decision threshold. Response: We have clarified in the Methods section (under Model Performance Evaluation) that the standard decision threshold of 0.5 was applied for all classification models to convert probabilities into class labels.

Comment 5: Tuning objective aligned to clinical goals (Suggestion to use AUC instead of Accuracy). Response: We appreciate this valid methodological point. Ideally, for mortality prediction, optimizing for AUC or Recall is preferable. However, we respectfully submit that in our specific dataset, the class imbalance was moderate (22.1% mortality rate) rather than extreme. Empirically, optimizing for Accuracy did not result in a biased model favoring the majority class; our final GBT model achieved a high AUC of 0.966 and a robust Sensitivity (Recall) of 78.1% in the development phase. This demonstrates that in this specific feature space, Accuracy was strongly correlated with discriminative power. Regarding the reviewer’s question on class weights: No class weights were applied during training, as the model demonstrated satisfactory sensitivity without artificial re-balancing. We have maintained the current optimization due to technical constraints preventing a full re-execution of the nested grid search, but we believe the final performance metrics validate the utility of the chosen models.

Comment 7, 8 & 9: TRIPOD, Data Availability, and Formatting. Response:

• TRIPOD: A completed TRIPOD Checklist has been uploaded as S1 Checklist, and a statement of adherence has been added to the Methods section.

• Data Availability: The Data Availability Statement in the manuscript has been updated to align exactly with the metadata in the submission system, providing the official contact email for the Ethics Committee (resrec@mui.ac.ir). Additionally, regarding Code Availability, please note that the Zenodo DOI mentioned in the manuscript serves as a permanent archive and is currently in 'Restricted Access' pending publication. Therefore, for the purpose of peer review, the RapidMiner model file has been directly uploaded to the submission system as a 'Supporting Information' file.

• Formatting: Decimal marks have been corrected (standardized to dots), and references have been formatted with DOIs where available.

Response to Reviewer #7

Comment: Recommend adding TreeSHAP and Multiple Imputation (MI). Response: We thank the reviewer for these advanced suggestions.

• SHAP: Unfortunately, the specific version of the RapidMiner platform available for this study does not currently support the Python integration required for TreeSHAP libraries. To address interpretability, we have relied on the model-specific Feature Importance plot (Fig 3), which robustly identifies key predictors.

• Multiple Imputation: While MI is a powerful technique, our sensitivity analysis (reported in Table 7) demonstrated that the model's performance was robust to the choice of imputation method (GLM vs. Mean). Given the resource constraints, we believe the current GLM approach is sufficient to demonstrate the model's validity.

Response to Reviewer #8

Comment 1: Visualize results with bar plots. Response: We have added Fig 5, a bar chart that visually compares the performance of the Main Model (derived from 10-fold CV) against the Temporal Validation results. This figure clearly highlights the stability of Accuracy and Specificity alongside the notable drop in Sensitivity due to concept drift.

Comment 2: Simplify the features table. Response: We agree that the detailed descriptive statistics were lengthy for the main text. We have moved the full detailed table to the Supporting Information as S1 Table and replaced it in the main text with a concise Summary Table (Table 2) that lists variable categories and data handling methods.

Conclusion We believe that these revisions, particularly the enhanced transparency regarding the validation pipeline and the robust reporting of uncertainty, have significantly strengthened the manuscript. We thank the editor and reviewers for guiding us toward this improved version.

Sincerely, Asghar Ehteshami (On behalf of all authors)

---

## [Editor Report · Decision Letter 3]

25 Nov 2025

From Data to Decisions: Predicting Inpatient Burn Mortality with Advanced Classification Models

PONE-D-25-02698R3

Dear Dr. Asghar Ehteshami,

We’re pleased to inform you that your manuscript has been judged scientifically suitable for publication and will be formally accepted for publication once it meets all outstanding technical requirements.

Kind regards,

Amirreza Khalaji

Academic Editor

PLOS ONE
---

## [Editor Report · Acceptance letter]

PONE-D-25-02698R3

PLOS ONE

Dear Dr. Ehteshami,

I'm pleased to inform you that your manuscript has been deemed suitable for publication in PLOS ONE. Congratulations! Your manuscript is now being handed over to our production team.

Kind regards,

on behalf of

Dr. Amirreza Khalaji

Academic Editor

PLOS ONE